# Active Learning from Imperfect Labelers

**Songbai Yan**
University of California, San Diego
yansongbai@eng.ucsd.edu

**Kamalika Chaudhuri**
University of California, San Diego
kamalika@cs.ucsd.edu

**Tara Javidi**
University of California, San Diego
tjavidi@eng.ucsd.edu

## Abstract

We study active learning where the labeler can not only return incorrect labels but also abstain from labeling. We consider different noise and abstention conditions of the labeler. We propose an algorithm which utilizes abstention responses, and analyze its statistical consistency and query complexity under fairly natural assumptions on the noise and abstention rate of the labeler. This algorithm is adaptive in a sense that it can automatically request less queries with a more informed or less noisy labeler. We couple our algorithm with lower bounds to show that under some technical conditions, it achieves nearly optimal query complexity.

## 1 Introduction

In active learning, the learner is given an input space $\mathcal{X}$, a label space $\mathcal{L}$, and a hypothesis class $\mathcal{H}$ such that one of the hypotheses in the class generates ground truth labels. Additionally, the learner has at its disposal a labeler to which it can pose interactive queries about the labels of examples in the input space. Note that the labeler may output a noisy version of the ground truth label (a flipped label). The goal of the learner is to learn a hypothesis in $\mathcal{H}$ which is close to the hypothesis that generates the ground truth labels.

There has been a significant amount of literature on active learning, both theoretical and practical. Previous theoretical work on active learning has mostly focused on the above basic setting [2, 4, 7, 10, 25] and has developed algorithms under a number of different models of label noise. A handful of exceptions include [3] which allows class conditional queries, [5] which allows requesting counterexamples to current version spaces, and [23, 26] where the learner has access to a strong labeler and one or more weak labelers.

In this paper, we consider a more general setting where, in addition to providing a possibly noisy label, the labeler can sometimes abstain from labeling. This scenario arises naturally in difficult labeling tasks and has been considered in computer vision by [11, 15]. Our goal in this paper is to investigate this problem from a foundational perspective, and explore what kind of conditions are needed, and how an abstaining labeler can affect properties such as consistency and query complexity of active learning algorithms.

The setting of active learning with an abstaining noisy labeler was first considered by [24], who looked at learning binary threshold classifiers based on queries to an labeler whose abstention rate is higher closer to the decision boundary. They primarily looked at the case when the abstention rate at a distance $\Delta$ from the decision boundary is less than $1 - \Theta(\Delta^\alpha)$, and the rate of label flips at the same distance is less than $\frac{1}{2} - \Theta(\Delta^\beta)$; under these conditions, they provided an active learning algorithm that given parameters $\alpha$ and $\beta$, outputs a classifier with error $\epsilon$ using $\tilde{O}(\epsilon^{-\alpha-2\beta})$ queries to the labeler.

However, there are several limitations to this work. The primary limitation is that parameters $\alpha$ and $\beta$ need to be known to the algorithm, which is not usually the case in practice. A second major limitation is that even if the labeler has nice properties, such as, the abstention rates increase sharply close to the boundary, their algorithm is unable to exploit these properties to reduce the number of queries. A third and final limitation is that their analysis only applies to one dimensional thresholds, and not to more general decision boundaries.

In this work, we provide an algorithm which is able to exploit nice properties of the labeler. Our algorithm is statistically consistent under very mild conditions — when the abstention rate is non-decreasing as we get closer to the decision boundary. Under slightly stronger conditions as in [24], our algorithm has the same query complexity. However, if the abstention rate of the labeler increases strictly monotonically close to the decision boundary, then our algorithm adapts and does substantially better. It simply exploits the increasing abstention rate close to the decision boundary, and does not even have to rely on the noisy labels! Specifically, when applied to the case where the noise rate is at most $\frac{1}{2} - \Theta(\Delta^\beta)$ and the abstention rate is $1 - \Theta(\Delta^\alpha)$ at distance $\Delta$ from the decision boundary, our algorithm can output a classifier with error $\epsilon$ based on only $\tilde{O}(\epsilon^{-\alpha})$ queries.

An important property of our algorithm is that the improvement of query complexity is achieved in a *completely adaptive manner*; unlike previous work [24], our algorithm needs *no information whatsoever on the abstention rates or rates of label noise*. Thus our result also strengthens existing results on active learning from (non-abstaining) noisy labelers by providing an adaptive algorithm that achieves that same performance as [6] without knowledge of noise parameters.

We extend our algorithm so that it applies to any smooth $d$-dimensional decision boundary in a non-parametric setting, not just one-dimensional thresholds, and we complement it with lower bounds on the number of queries that need to be made to any labeler. Our lower bounds generalize the lower bounds in [24], and shows that our upper bounds are nearly optimal. We also present an example that shows that at least a relaxed version of the monotonicity property is necessary to achieve this performance gain; if the abstention rate plateaus around the decision boundary, then our algorithm needs to query and rely on the noisy labels (resulting in higher query complexity) in order to find a hypothesis close to the one generating the ground truth labels.

## 1.1 Related work

There has been a considerable amount of work on active learning, most of which involves labelers that are not allowed to abstain. Theoretical work on this topic largely falls under two categories — the membership query model [6, 13, 18, 19], where the learner can request label of any example in the instance space, and the PAC model, where the learner is given a large set of unlabeled examples from an underlying unlabeled data distribution, and can request labels of a subset of these examples. Our work and also that of [24] builds on the membership query model.

There has also been a lot of work on active learning under different noise models. The problem is relatively easy when the labeler always provides the ground truth labels – see [8, 9, 12] for work in this setting in the PAC model, and [13] for the membership query model. Perhaps the simplest setting of label noise is random classification noise, where each label is flipped with a probability that is independent of the unlabeled instance. [14] shows how to address this kind of noise in the PAC model by repeatedly querying an example until the learner is confident of its label; [18, 19] provide more sophisticated algorithms with better query complexities in the membership query model. A second setting is when the noise rate increases closer to the decision boundary; this setting has been studied under the membership query model by [6] and in the PAC model by [10, 4, 25]. A final setting is agnostic PAC learning — when a fixed but arbitrary fraction of labels may disagree with the label assigned by the optimal hypothesis in the hypothesis class. Active learning is known to be particularly difficult in this setting; however, algorithms and associated label complexity bounds have been provided by [1, 2, 4, 10, 12, 25] among others.

Our work expands on the membership query model, and our abstention and noise models are related to a variant of the Tsybakov noise condition. A setting similar to ours was considered by [6, 24]. [6] considers a non-abstaining labeler, and provides a near-optimal binary search style active learning algorithm; however, their algorithm is non-adaptive. [24] gives a nearly matching lower and upper query complexity bounds for active learning with abstention feedback, but they only give a non-adaptive algorithm for learning one dimensional thresholds, and only study the situation where the

abstention rate is upper-bounded by a polynomial function. Besides [24] , [11, 15] study active learning with abstention feedback in computer vision applications. However, these works are based on heuristics and do not provide any theoretical guarantees.

## 2 Settings

**Notation.** $\mathbb{1}[A]$ *is the indicator function:* $\mathbb{1}[A] = 1$ *if $A$ is true, and 0 otherwise. For* $\boldsymbol{x} = (x_1, \ldots, x_d) \in \mathbb{R}^d$ *(*$d > 1$*), denote* $(x_1, \ldots, x_{d-1})$ *by* $\tilde{\boldsymbol{x}}$*. Define* $\ln x = \log_e x$*,* $\log x = \log_{\frac{4}{3}} x$*,* $[\ln \ln]_+ (x) = \ln \ln \max\{x, e^e\}$*. We use* $\tilde{O}$ *and* $\tilde{\Theta}$ *to hide logarithmic factors in* $\frac{1}{\epsilon}$*,* $\frac{1}{\delta}$*, and d.*

**Definition.** *Suppose* $\gamma \geq 1$*. A function* $g : [0,1]^{d-1} \rightarrow \mathbb{R}$ *is* $(K, \gamma)$*-Hölder smooth, if it is continuously differentiable up to* $\lfloor \gamma \rfloor$*-th order, and for any* $\boldsymbol{x}, \boldsymbol{y} \in [0,1]^{d-1}$*,* $\left| g(\boldsymbol{y}) - \sum_{m=0}^{\lfloor \gamma \rfloor} \frac{\partial^m g(\boldsymbol{x})}{m!} (\boldsymbol{y} - \boldsymbol{x})^m \right| \leq K \|\boldsymbol{y} - \boldsymbol{x}\|^\gamma$*. We denote this class of functions by* $\Sigma(K, \gamma)$*.*

We consider active learning for binary classification. We are given an instance space $\mathcal{X} = [0,1]^d$ and a label space $\mathcal{L} = \{0, 1\}$. Each instance $x \in \mathcal{X}$ is assigned to a label $l \in \{0, 1\}$ by an underlying function $h^* : \mathcal{X} \rightarrow \{0, 1\}$ unknown to the learning algorithm in a hypothesis space $\mathcal{H}$ of interest. The learning algorithm has access to any $x \in \mathcal{X}$, but no access to their labels. Instead, it can only obtain label information through interactions with a labeler, whose relation to $h^*$ is to be specified later. The objective of the algorithm is to sequentially select the instances to query for label information and output a classifier $\hat{h}$ that is close to $h^*$ while making as few queries as possible.

We consider a non-parametric setting as in [6, 17] where the hypothesis space is the *smooth boundary fragment* class $\mathcal{H} = \{h_g(\boldsymbol{x}) = \mathbb{1}[x_d > g(\tilde{\boldsymbol{x}})] \mid g : [0,1]^{d-1} \rightarrow [0,1] \text{ is } (K, \gamma)\text{-Hölder smooth}\}$. In other words, the decision boundaries of classifiers in this class are epigraph of smooth functions (see Figure 1 for example). We assume $h^*(\boldsymbol{x}) = \mathbb{1}[x_d > g^*(\tilde{\boldsymbol{x}})] \in \mathcal{H}$. When $d = 1$, $\mathcal{H}$ reduces to the space of threshold functions $\{h_\theta(x) = \mathbb{1}[x > \theta] : \theta \in [0,1]\}$.

The performance of a classifier $h(\boldsymbol{x}) = \mathbb{1}[x_d > g(\tilde{\boldsymbol{x}})]$ is evaluated by the $L^1$ distance between the decision boundaries $\|g - g^*\| = \int_{[0,1]^{d-1}} |g(\tilde{\boldsymbol{x}}) - g^*(\tilde{\boldsymbol{x}})| \, d\tilde{\boldsymbol{x}}$.

The learning algorithm can only obtain label information by querying a labeler who is allowed to abstain from labeling or return an incorrect label (flipping between 0 and 1). For each query $\boldsymbol{x} \in [0,1]^d$, the labeler $L$ will return $y \in \mathcal{Y} = \{0, 1, \perp\}$ ($\perp$ means that the labeler abstains from providing a 0/1 label) according to some distribution $P_L(Y = y \mid X = \boldsymbol{x})$. When it is clear from the context, we will drop the subscript from $P_L(Y \mid X)$. Note that while the labeler can declare its indecision by outputting $\perp$, we do not allow classifiers in our hypothesis space to output $\perp$.

In our active learning setting, our goal is to output a boundary $g$ that is close to $g^*$ while making as few interactive queries to the labeler as possible. In particular, we want to find an algorithm with low *query complexity* $\Lambda(\epsilon, \delta, \mathcal{A}, L, g^*)$, which is defined as the minimum number of queries that Algorithm $\mathcal{A}$, acting on samples with ground truth $g^*$, should make to a labeler $L$ to ensure that the output classifier $h_g(\boldsymbol{x}) = \mathbb{1}[x_d > g(\tilde{\boldsymbol{x}})]$ has the property $\|g - g^*\| = \int_{[0,1]^{d-1}} |g(\tilde{\boldsymbol{x}}) - g^*(\tilde{\boldsymbol{x}})| \, d\tilde{\boldsymbol{x}} \leq \epsilon$ with probability at least $1 - \delta$ over the responses of $L$.

### 2.1 Conditions

We now introduce three conditions on the response of the labeler with increasing strictness. Later we will provide an algorithm whose query complexity improves with increasing strictness of conditions.

**Condition 1.** *The response distribution of the labeler* $P(Y \mid X)$ *satisfies:*

- *(abstention) For any* $\tilde{\boldsymbol{x}} \in [0,1]^{d-1}$*,* $x_d, x_d' \in [0,1]$*, if* $|x_d - g^*(\tilde{\boldsymbol{x}})| \geq |x_d' - g^*(\tilde{\boldsymbol{x}})|$ *then* $P(\perp \mid (\tilde{\boldsymbol{x}}, x_d)) \leq P(\perp \mid (\tilde{\boldsymbol{x}}, x_d'))$*;*

- *(noise) For any* $\boldsymbol{x} \in [0,1]^d$*,* $P(Y \neq \mathbb{1}[x_d > g^*(\tilde{\boldsymbol{x}})] \mid \boldsymbol{x}, Y \neq \perp) \leq \frac{1}{2}$*.*

Condition 1 means that the closer $\boldsymbol{x}$ is to the decision boundary $(\tilde{\boldsymbol{x}}, g^*(\tilde{\boldsymbol{x}}))$, the more likely the labeler is to abstain from labeling. This complies with the intuition that instances closer to the decision boundary are harder to classify. We also assume the 0/1 labels can be flipped with probability as large as $\frac{1}{2}$. In other words, we allow unbounded noise.

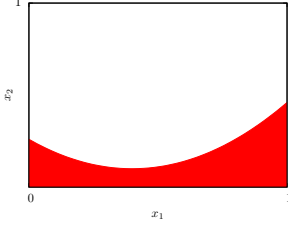

Figure 1: A classifier with boundary $g(\tilde{\boldsymbol{x}}) = (x_1 - 0.4)^2 + 0.1$ for $d = 2$. Label 1 is assigned to the region above, 0 to the below (red region)

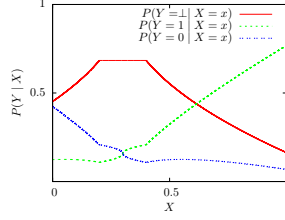

Figure 2: The distributions above satisfy Conditions 1 and 2, but the abstention feedback is useless since $P(\bot \mid x)$ is flat between $x = 0.2$ and 0.4

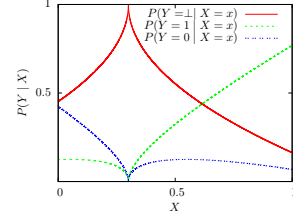

Figure 3: Distributions above satisfy Conditions 1, 2, and 3.

**Condition 2.** *Let $C, \beta$ be non-negative constants, and $f : [0, 1] \to [0, 1]$ be a nondecreasing function. The response distribution $P(Y \mid X)$ satisfies:*

- *(abstention)* $P(\bot \mid \boldsymbol{x}) \leq 1 - f\left(|x_d - g^*(\tilde{\boldsymbol{x}})|\right)$;

- *(noise)* $P(Y \neq \mathbb{1}\left[x_d > g^*(\tilde{\boldsymbol{x}})\right] \mid \boldsymbol{x}, Y \neq \bot) \leq \frac{1}{2}\left(1 - C\left|x_d - g^*(\tilde{\boldsymbol{x}})\right|^{\beta}\right)$.

Condition 2 requires the abstention and noise probabilities to be upper-bounded, and these upper bounds decrease as $\boldsymbol{x}$ moves further away from the decision boundary. The abstention rate can be 1 at the decision boundary, so the labeler may always abstain at the decision boundary. The condition on the noise satisfies the popular Tsybakov noise condition [22].

**Condition 3.** *Let $f : [0, 1] \to [0, 1]$ be a nondecreasing function such that $\exists 0 < c < 1$, $\forall 0 < a \leq 1$ $\forall 0 \leq b \leq \frac{2}{3}a$, $\frac{f(b)}{f(a)} \leq 1 - c$. The response distribution satisfies: $P(\bot \mid \boldsymbol{x}) = 1 - f\left(|x_d - g^*(\tilde{\boldsymbol{x}})|\right)$.*

An example where Condition 3 holds is $P(\bot \mid \boldsymbol{x}) = 1 - (x - 0.3)^{\alpha}$ ($\alpha > 0$).

Condition 3 requires the abstention rate to increase monotonically close to the decision boundary as in Condition 1. In addition, it requires the abstention probability $P(\bot \mid (\tilde{\boldsymbol{x}}, x_d))$ not to be too flat with respect to $x_d$. For example, when $d = 1$, $P(\bot \mid x) = 0.68$ for $0.2 \leq x \leq 0.4$ (shown as Figure 2) does not satisfy Condition 3, and abstention responses are not informative since this abstention rate alone yields no information on the location of the decision boundary. In contrast, $P(\bot \mid x) = 1 - \sqrt{|x - 0.3|}$ (shown as Figure 3) satisfies Condition 3, and the learner could infer it is getting close to the decision boundary when it starts receiving more abstention responses.

Note that here $c, f, C, \beta$ are *unknown* and *arbitrary* parameters that characterize the complexity of the learning task. We want to design an algorithm that does not require knowledge of these parameters but still achieves nearly optimal query complexity.

## 3 Learning one-dimensional thresholds

In this section, we start with the one dimensional case ($d = 1$) to demonstrate the main idea. We will generalize these results to multidimensional instance space in the next section.

When $d = 1$, the decision boundary $g^*$ becomes a point in $[0, 1]$, and the corresponding classifier is a threshold function over [0,1]. In other words the hypothesis space becomes $\mathcal{H} = \{f_\theta(x) = \mathbb{1}\left[x > \theta\right] : \theta \in [0, 1]\}$). We denote the ground truth decision boundary by $\theta^* \in [0, 1]$. We want to find a $\hat{\theta} \in [0, 1]$ such that $|\hat{\theta} - \theta^*|$ is small while making as few queries as possible.

### 3.1 Algorithm

The proposed algorithm is a binary search style algorithm shown as Algorithm 1. (For the sake of simplicity, we assume $\log \frac{1}{2\epsilon}$ is an integer.) Algorithm 1 takes a desired precision $\epsilon$ and confidence

**Algorithm 1** The active learning algorithm for learning thresholds

1: Input: $\delta, \epsilon$
2: $[L_0, R_0] \leftarrow [0, 1]$
3: **for** $k = 0, 1, 2, \ldots, \log \frac{1}{2\epsilon} - 1$ **do**
4:      Define three quartiles: $U_k \leftarrow \frac{3L_k + R_k}{4}$, $M_k \leftarrow \frac{L_k + R_k}{2}$, $V_k \leftarrow \frac{L_k + 3R_k}{4}$
5:      $A^{(u)}, A^{(m)}, A^{(v)}, B^{(u)}, B^{(v)} \leftarrow$ Empty Array
6:      **for** $n = 1, 2, \ldots$ **do**
7:          Query at $U_k, M_k, V_k$, and receive labels $X_n^{(u)}, X_n^{(m)}, X_n^{(v)}$
8:          **for** $w \in \{u, m, v\}$ **do**
9:              ▷ We record whether $X^{(w)} = \perp$ in $A^{(w)}$, and the 0/1 label (as -1/1) in $B^{(w)}$ if $X^{(w)} \neq \perp$
10:              **if** $X^{(w)} \neq \perp$ **then**
11:                  $A^{(w)} \leftarrow A^{(w)}$.append(1) , $B^{(w)} \leftarrow B^{(w)}$.append($2\mathbb{1}\left[X^{(w)} = 1\right] - 1$)
12:              **else**
13:                  $A^{(w)} \leftarrow A^{(w)}$.append(0)
14:              **end if**
15:          **end for**
16:          ▷ Check if the differences of abstention responses are statistically significant
17:          **if** CHECKSIGNIFICANT-VAR$\left(\left\{ A_i^{(u)} - A_i^{(m)} \right\}_{i=1}^n, \frac{\delta}{4 \log \frac{1}{2\epsilon}}\right)$ **then**
18:              $[L_{k+1}, R_{k+1}] \leftarrow [U_k, R_k]$; break
19:          **else if** CHECKSIGNIFICANT-VAR$\left(\left\{ A_i^{(v)} - A_i^{(m)} \right\}_{i=1}^n, \frac{\delta}{4 \log \frac{1}{2\epsilon}}\right)$ **then**
20:              $[L_{k+1}, R_{k+1}] \leftarrow [L_k, V_k]$; break
21:          **end if**
22:          ▷ Check if the differences between 0 and 1 labels are statistically significant
23:          **if** CHECKSIGNIFICANT$\left(\left\{ -B_i^{(u)} \right\}_{i=1}^{B^{(u)}.\text{length}}, \frac{\delta}{4 \log \frac{1}{2\epsilon}}\right)$ **then**
24:              $[L_{k+1}, R_{k+1}] \leftarrow [U_k, R_k]$; break
25:          **else if** CHECKSIGNIFICANT$\left(\left\{ B_i^{(v)} \right\}_{i=1}^{B^{(v)}.\text{length}}, \frac{\delta}{4 \log \frac{1}{2\epsilon}}\right)$ **then**
26:              $[L_{k+1}, R_{k+1}] \leftarrow [L_k, V_k]$; break
27:          **end if**
28:      **end for**
29: **end for**
30: Output: $\hat{\theta} = \left( L_{\log \frac{1}{2\epsilon}} + R_{\log \frac{1}{2\epsilon}} \right) / 2$

---

level $\delta$ as its input, and returns an estimation $\hat{\theta}$ of the decision boundary $\theta^*$. The algorithm maintains an interval $[L_k, R_k]$ in which $\theta^*$ is believed to lie, and shrinks this interval iteratively. To find the subinterval that contains $\theta^*$, Algorithm 1 relies on two auxiliary functions (marked in Procedure 2) to conduct adaptive sequential hypothesis tests regarding subintervals of interval $[L_k, R_k]$.

Suppose $\theta^* \in [L_k, R_k]$. Algorithm 1 tries to shrink this interval to a $\frac{3}{4}$ of its length in each iteration by repetitively querying on quartiles $U_k = \frac{3L_k + R_k}{4}$, $M_k = \frac{L_k + R_k}{2}$, $V_k = \frac{L_k + 3R_k}{4}$. To determine which specific subinterval to choose, the algorithm uses 0/1 labels and abstention responses simultaneously. Since the ground truth labels are determined by $\mathbb{1}\left[x > \theta^*\right]$, one can infer that if the number of queries that return label 0 at $U_k$ ($V_k$) is statistically significantly more (less) than label 1, then $\theta^*$ should be on the right (left) side of $U_k$ ($V_k$). Similarly, from Condition 1, if the number of non-abstention responses at $U_k$ ($V_k$) is statistically significantly more than non-abstention responses at $M_k$, then $\theta^*$ should be closer to $M_k$ than $U_k$ ($V_k$).

Algorithm 1 relies on the ability to shrink the search interval via statistically comparing the numbers of obtained labels at locations $U_k, M_k, V_k$. As a result, a main building block of Algorithm 1 is to test whether i.i.d. bounded random variables $Y_i$ are greater in expectation than i.i.d. bounded random variables $Z_i$ with statistical significance. In Procedure 2, we have two test functions CheckSignificant and CheckSignificant-Var that take i.i.d. random variables $\{X_i = Y_i - Z_i\}$ ($|X_i| \leq 1$) and confidence level $\delta$ as their input, and output whether it is statistically significant to conclude $\mathbb{E}X_i > 0$.

---

**Procedure 2** Adaptive sequential testing

1: ▷ $D_0, D_1$ are absolute constants defined in Proposition 1 and Proposition 2
2: ▷ $\{X_i\}$ are i.i.d. random variables bounded by 1. $\delta$ is the confidence level. Detect if $\mathbb{E}X > 0$
3: **function** CHECKSIGNIFICANT($\{X_i\}_{i=1}^{n}, \delta$)
4:      $p(n, \delta) \leftarrow D_0 \left( 1 + \ln \frac{1}{\delta} + \sqrt{4n \left( [\ln \ln]_+ 4n + \ln \frac{1}{\delta} \right)} \right)$
5:      Return $\sum_{i=1}^{n} X_i \geq p(n, \delta)$
6: **end function**
7: **function** CHECKSIGNIFICANT-VAR($\{X_i\}_{i=1}^{n}, \delta$)
8:      Calculate the empirical variance Var $= \frac{n}{n-1} \left( \sum_{i=1}^{n} X_i^2 - \frac{1}{n} \left( \sum_{i=1}^{n} X_i \right)^2 \right)$
9:      $q(n, \text{Var}, \delta) \leftarrow D_1 \left( 1 + \ln \frac{1}{\delta} + \sqrt{\left( \text{Var} + \ln \frac{1}{\delta} + 1 \right) \left( [\ln \ln]_+ \left( \text{Var} + \ln \frac{1}{\delta} + 1 \right) + \ln \frac{1}{\delta} \right)} \right)$
10:      Return $n \geq \ln \frac{1}{\delta}$ AND $\sum_{i=1}^{n} X_i \geq q(n, \text{Var}, \delta)$
11: **end function**

---

CheckSignificant is based on the following uniform concentration result regarding the empirical mean:

**Proposition 1.** *Suppose $X_1, X_2, \ldots$ are a sequence of i.i.d. random variables with $X_1 \in [-2, 2]$, $\mathbb{E}X_1 = 0$. Take any $0 < \delta < 1$. Then there is an absolute constant $D_0$ such that with probability at least $1 - \delta$, for all $n > 0$ simultaneously,*

$$\left| \sum_{i=1}^{n} X_i \right| \leq D_0 \left( 1 + \ln \frac{1}{\delta} + \sqrt{4n \left( [\ln \ln]_+ 4n + \ln \frac{1}{\delta} \right)} \right)$$

In Algorithm 1, we use CheckSignificant to detect whether the expected number of queries that return label 0 at location $U_k$ ($V_k$) is more/less than the expected number of label 1 with a statistical significance.

CheckSignificant-Var is based on the following uniform concentration result which further utilizes the empirical variance $V_n = \frac{n}{n-1} \left( \sum_{i=1}^{n} X_i^2 - \frac{1}{n} \left( \sum_{i=1}^{n} X_i \right)^2 \right)$:

**Proposition 2.** *There is an absolute constant $D_1$ such that with probability at least $1 - \delta$, for all $n \geq \ln \frac{1}{\delta}$ simultaneously,*

$$\left| \sum_{i=1}^{n} X_i \right| \leq D_1 \left( 1 + \ln \frac{1}{\delta} + \sqrt{\left( 1 + \ln \frac{1}{\delta} + V_n \right) \left( [\ln \ln]_+ \left( 1 + \ln \frac{1}{\delta} + V_n \right) + \ln \frac{1}{\delta} \right)} \right)$$

The use of variance results in a tighter bound when $\text{Var}(X_i)$ is small.

In Algorithm 1, we use CheckSignificant-Var to detect the statistical significance of the relative order of the number of queries that return non-abstention responses at $U_k$ ($V_k$) compared to the number of non-abstention responses at $M_k$. This results in a better query complexity than using CheckSignificant under Condition 3, since the variance of the number of abstention responses approaches 0 when the interval $[L_k, R_k]$ zooms in on $\theta^*$.[1]

## 3.2 Analysis

For Algorithm 1 to be statistically consistent, we only need Condition 1.

**Theorem 1.** *Let $\theta^*$ be the ground truth. If the labeler $L$ satisfies Condition 1 and Algorithm 1 stops to output $\hat{\theta}$, then $\left| \theta^* - \hat{\theta} \right| \leq \epsilon$ with probability at least $1 - \frac{\delta}{2}$.*

Under additional Conditions 2 and 3, we can derive upper bounds of the query complexity for our algorithm. (Recall $f$ and $\beta$ are defined in Conditions 2 and 3.)

**Theorem 2.** *Let $\theta^*$ be the ground truth, and $\hat{\theta}$ be the output of Algorithm 1. Under Conditions 1 and 2, with probability at least $1 - \delta$, Algorithm 1 makes at most $\tilde{O}\left(\frac{1}{f(\frac{\epsilon}{2})}\epsilon^{-2\beta}\right)$ queries.*

**Theorem 3.** *Let $\theta^*$ be the ground truth, and $\hat{\theta}$ be the output of Algorithm 1. Under Conditions 1 and 3, with probability at least $1 - \delta$, Algorithm 1 makes at most $\tilde{O}\left(\frac{1}{f(\frac{\epsilon}{2})}\right)$ queries.*

The query complexity given by Theorem 3 is independent of $\beta$ that decides the flipping rate, and consequently smaller than the bound in Theorem 2. This improvement is due to the use of abstention responses, which become much more informative under Condition 3.

### 3.3 Lower Bounds

In this subsection, we give lower bounds of query complexity in the one-dimensional case and establish near optimality of Algorithm 1. We will give corresponding lower bounds for the high-dimensional case in the next section.

The lower bound in [24] can be easily generalized to Condition 2:

**Theorem 4.** *([24]) There is a universal constant $\delta_0 \in (0, 1)$ and a labeler $L$ satisfying Conditions 1 and 2, such that for any active learning algorithm $\mathcal{A}$, there is a $\theta^* \in [0, 1]$, such that for small enough $\epsilon$, $\Lambda(\epsilon, \delta_0, \mathcal{A}, L, \theta^*) \geq \Omega\left(\frac{1}{f(\epsilon)}\epsilon^{-2\beta}\right)$.*

Our query complexity (Theorem 3) for the algorithm is also almost tight under Conditions 1 and 3 with a polynomial abstention rate.

**Theorem 5.** *There is a universal constant $\delta_0 \in (0, 1)$ and a labeler $L$ satisfying Conditions 1, 2, and 3 with $f(x) = C'x^\alpha$ ($C' > 0$ and $0 < \alpha \leq 2$ are constants), such that for any active learning algorithm $\mathcal{A}$, there is a $\theta^* \in [0, 1]$, such that for small enough $\epsilon$, $\Lambda(\epsilon, \delta_0, \mathcal{A}, L, \theta^*) \geq \Omega\left(\epsilon^{-\alpha}\right)$.*

### 3.4 Remarks

Our results confirm the intuition that learning with abstention is easier than learning with noisy labels. This is true because a noisy label might mislead the learning algorithm, but an abstention response never does. Our analysis shows, in particular, that if the labeler never abstains, and outputs completely noisy labels with probability bounded by $1 - |x - \theta^*|^\gamma$ (i.e., $P(Y \neq \mathbb{I}[x > \theta^*] \mid x) \leq \frac{1}{2}(1 - |x - \theta^*|^\gamma)$), then the near optimal query complexity of $\tilde{O}\left(\epsilon^{-2\gamma}\right)$ is significantly larger than the near optimal $\tilde{O}\left(\epsilon^{-\gamma}\right)$ query complexity associated with a labeler who only abstains with probability $P(Y = \perp \mid x) \leq 1 - |x - \theta^*|^\gamma$ and never flips a label. More precisely, while in both cases the labeler outputs the same amount of corrupted labels, the query complexity of the abstention-only case is significantly smaller than the noise-only case.

Note that the query complexity of Algorithm 1 consists of two kinds of queries: queries which return 0/1 labels and are used by function CheckSignificant, and queries which return abstention and are used by function CheckSignificant-Var. Algorithm 1 will stop querying when the responses of one of the two kinds of queries are statistically significant. Under Condition 2, our proof actually shows that the optimal number of queries is dominated by the number of queries used by CheckSignificant function. In other words, a simplified variant of Algorithm 1 which excludes use of abstention feedback is near optimal. Similarly, under Condition 3, the optimal query complexity is dominated by the number of queries used by CheckSignificant-Var function. Hence the variant of Algorithm 1 which disregards 0/1 labels would be near optimal.

## 4 The multidimensional case

We follow [6] to generalize the results from one-dimensional thresholds to the d-dimensional ($d > 1$) smooth boundary fragment class $\Sigma(K, \gamma)$.

---

**Algorithm 3** The active learning algorithm for the smooth boundary fragment class

---

1: Input: $\delta, \epsilon, \gamma$
2: $M \leftarrow \Theta\left(\epsilon^{-1/\gamma}\right)$. $\mathcal{L} \leftarrow \left\{\frac{0}{M}, \frac{1}{M}, \ldots, \frac{M-1}{M}\right\}^{d-1}$
3: For each $l \in \mathcal{L}$, apply Algorithm 1 with parameter $(\epsilon, \delta/M^{d-1})$ to learn a threshold $g_l$ that approximates $g^*(l)$
4: Partition the instance space into cells $\{I_q\}$ indexed by $q \in \left\{0, 1, \ldots, \frac{M}{\gamma} - 1\right\}^{d-1}$, where

$$I_q = \left[\frac{q_1 \gamma}{M}, \frac{(q_1+1)\gamma}{M}\right] \times \cdots \times \left[\frac{q_{d-1}\gamma}{M}, \frac{(q_{d-1}+1)\gamma}{M}\right]$$

5: For each cell $I_q$, perform a polynomial interpolation: $g_q(\tilde{\boldsymbol{x}}) = \sum_{l \in I_q \cap \mathcal{L}} g_l Q_{q,l}(\tilde{\boldsymbol{x}})$, where

$$Q_{q,l}(\tilde{\boldsymbol{x}}) = \prod_{i=1}^{d-1} \prod_{j=0, j \neq Ml_i - \gamma q_i}^{\gamma} \frac{\tilde{\boldsymbol{x}}_i - (\gamma q_i + j)/M}{l_i - (\gamma q_i + j)/M}$$

6: Output: $g(\tilde{\boldsymbol{x}}) = \sum_{q \in \left\{0, 1, \ldots, \frac{M}{\gamma} - 1\right\}^{d-1}} g_q(\tilde{\boldsymbol{x}}) \mathbb{1}\left[\tilde{\boldsymbol{x}} \in q\right]$

---

### 4.1 Lower bounds

**Theorem 6.** *There are universal constants $\delta_0 \in (0, 1)$, $c_0 > 0$, and a labeler $L$ satisfying Conditions 1 and 2, such that for any active learning algorithm $\mathcal{A}$, there is a $g^* \in \Sigma(K, \gamma)$, such that for small enough $\epsilon$, $\Lambda(\epsilon, \delta_0, \mathcal{A}, L, g^*) \geq \Omega\left(\frac{1}{f(c_0\epsilon)}\epsilon^{-2\beta - \frac{d-1}{\gamma}}\right)$.*

**Theorem 7.** *There is a universal constant $\delta_0 \in (0, 1)$ and a labeler $L$ satisfying Conditions 1, 2, and Condition 3 with $f(x) = C'x^\alpha$ ($C' > 0$ and $0 < \alpha \leq 2$ are constants), such that for any active learning algorithm $\mathcal{A}$, there is a $g^* \in \Sigma(K, \gamma)$, such that for small enough $\epsilon$, $\Lambda(\epsilon, \delta_0, \mathcal{A}, L, g^*) \geq \Omega\left(\epsilon^{-\alpha - \frac{d-1}{\gamma}}\right)$.*

### 4.2 Algorithm and Analysis

Recall the decision boundary of the smooth boundary fragment class can be seen as the epigraph of a smooth function $[0, 1]^{d-1} \rightarrow [0, 1]$. For $d > 1$, we can reduce the problem to the one-dimensional problem by discretizing the first $d-1$ dimensions of the instance space and then perform a polynomial interpolation. The algorithm is shown as Algorithm 3. For the sake of simplicity, we assume $\gamma, M/\gamma$ in Algorithm 3 are integers.

We have similar consistency guarantee and upper bounds as in the one-dimensional case.

**Theorem 8.** *Let $g^*$ be the ground truth. If the labeler $L$ satisfies Condition 1 and Algorithm 3 stops to output $g$, then $\|g^* - g\| \leq \epsilon$ with probability at least $1 - \frac{\delta}{2}$.*

**Theorem 9.** *Let $g^*$ be the ground truth, and $g$ be the output of Algorithm 3. Under Conditions 1 and 2, with probability at least $1 - \delta$, Algorithm 3 makes at most $\tilde{O}\left(\frac{d}{f(\epsilon/2)}\epsilon^{-2\beta - \frac{d-1}{\gamma}}\right)$ queries.*

**Theorem 10.** *Let $g^*$ be the ground truth, and $g$ be the output of Algorithm 3. Under Conditions 1 and 3, with probability at least $1 - \delta$, Algorithm 3 makes at most $\tilde{O}\left(\frac{d}{f(\epsilon/2)}\epsilon^{-\frac{d-1}{\gamma}}\right)$ queries.*

**Acknowledgments.** We thank NSF under IIS-1162581, CCF-1513883, and CNS-1329819 for research support.

## Footnotes

[1]We do not apply CheckSignificant-Var to 0/1 labels, because unlike the difference between the numbers of abstention responses at $U_k$ ($V_k$) and $M_k$, the variance of the difference between the numbers of 0 and 1 labels stays above a positive constant.

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
