[Supplementary Material · supplementary.pdf]

## A Proof of query complexities

### A.1 Properties of adaptive sequential testing in Procedure 2

**Lemma 1.** *Suppose $\{X_i\}_{i=1}^{\infty}$ is a sequence of i.i.d. random variables such that $\mathbb{E}X_i \leq 0$, $|X_i| \leq 1$. Let $\delta > 0$. Then with probability at least $1 - \delta$, for all $n \in \mathbb{N}$ simultaneously CheckSignificant$(\{X_i\}_{i=1}^{n}, \delta)$ in Procedure 2 returns false.*

*Proof.* This is immediate by applying Proposition 1 to $X_i - \mathbb{E}X_i$. $\square$

**Lemma 2.** *Suppose $\{X_i\}_{i=1}^{\infty}$ is a sequence of i.i.d. random variables such that $\mathbb{E}X_i > \epsilon > 0$, $|X_i| \leq 1$. Let $\delta \in [0, \frac{1}{3}]$, $N \geq \frac{\xi}{\epsilon^2} \ln\frac{1}{\delta} [\ln\ln]_+ \frac{1}{\epsilon}$ ($\xi$ is an absolute constant specified in the proof). Then with probability at least $1 - \delta$, CheckSignificant$\left(\{X_i\}_{i=1}^{N}, \delta\right)$ in Procedure 2 returns true.*

*Proof.* Let $S_N = \sum_{i=1}^{N} X_i$. CheckSignificant$\left(\{X_i\}_{i=1}^{N}, \delta\right)$ returns false if and only if $S_N \leq D_0 \left(1 + \ln\frac{1}{\delta} + \sqrt{N\left([\ln\ln]_+ N + \ln\frac{1}{\delta}\right)}\right)$.

$$
\Pr\left(S_N \leq D_0\left(1 + \ln\frac{1}{\delta} + \sqrt{N\left([\ln\ln]_+ N + \ln\frac{1}{\delta}\right)}\right)\right)
$$
$$
\leq \Pr\left(S_N \leq D_0\left(1 + \ln\frac{1}{\delta} + \sqrt{N[\ln\ln]_+ N} + \sqrt{N\ln\frac{1}{\delta}}\right)\right)
$$
$$
\leq \Pr\left(S_N - N\mathbb{E}X_i \leq D_0\left(1 + \ln\frac{1}{\delta} + \sqrt{N[\ln\ln]_+ N} + \sqrt{N\ln\frac{1}{\delta}}\right) - N\epsilon\right)
$$

Suppose $N = \frac{c\xi}{\epsilon^2} \ln\frac{1}{\delta} [\ln\ln]_+ \frac{1}{\epsilon}$ for constant $c \geq 1$ and $\xi$. $\xi$ is set to be sufficiently large, such that (1) $\xi \geq 4D_0^2$; (2) $\frac{2D_0}{\sqrt{\xi}} + D_0\left(3 + \sqrt{[\ln\ln]_+ \xi}\right) + D_0 - \sqrt{\xi}/2 \leq -\sqrt{\frac{1}{2}}$; (3) $f(x) = D_0\sqrt{[\ln\ln]_+ x} - \sqrt{x}/2$ is decreasing when $x > \xi$. Here (2) is satisfiable since $\frac{D_0}{\sqrt{\xi}} + D_0\sqrt{[\ln\ln]_+ \xi} - \sqrt{\xi}/2 \to -\infty$ as $\xi \to \infty$, (3) is satisfiable since $f'(x) \to -\infty$ as $x \to \infty$. (2) and (3) together implies $\frac{2D_0}{\sqrt{\xi}} + D_0\left(3 + \sqrt{[\ln\ln]_+ c\xi}\right) + D_0 - \sqrt{c\xi}/2 \leq -\sqrt{\frac{1}{2}}$.

$$
\frac{1}{\sqrt{N}}\left(D_0\left(1 + \ln\frac{1}{\delta} + \sqrt{N[\ln\ln]_+ N} + \sqrt{N\ln\frac{1}{\delta}}\right) - N\epsilon\right)
$$
$$
= \sqrt{\ln\frac{1}{\delta}}\left(\frac{D_0\epsilon(1 + \ln\frac{1}{\delta})}{\sqrt{c\xi[\ln\ln]_+ \frac{1}{\epsilon}}\ln\frac{1}{\delta}} + D_0\sqrt{\frac{[\ln\ln]_+ \left(\frac{c\xi}{\epsilon^2}\ln\frac{1}{\delta}[\ln\ln]_+ \frac{1}{\epsilon}\right)}{\ln\frac{1}{\delta}}} + D_0 - \sqrt{c\xi[\ln\ln]_+ \frac{1}{\epsilon}}\right)
$$

Since $[\ln\ln]_+ \frac{1}{\epsilon}, c, \ln\frac{1}{\delta} \geq 1$ and $\epsilon < 1$, we have $\frac{D_0\epsilon(1 + \ln\frac{1}{\delta})}{\sqrt{c\xi[\ln\ln]_+ \frac{1}{\epsilon}}\ln\frac{1}{\delta}} \leq \frac{2D_0}{\sqrt{\xi}}$.

Since $[\ln\ln]_+ x \geq 1$ if $x \geq 1$, we have $[\ln\ln]_+ \frac{1}{\epsilon} \leq \frac{1}{\epsilon}$, and thus

$$\sqrt{[\ln\ln]_+ \left(\frac{c\xi}{\epsilon^2}\ln\frac{1}{\delta}[\ln\ln]_+ + \frac{1}{\epsilon}\right)} = \sqrt{\ln\left[\max\left\{e, 2\ln\frac{1}{\epsilon} + \ln c\xi + \ln\ln\frac{1}{\delta} + \ln[\ln\ln]_+ + \frac{1}{\epsilon}\right\}\right]}$$

$$\leq \sqrt{\ln\left[\max\left\{e, 3\ln\frac{1}{\epsilon} + \ln c\xi + [\ln\ln]_+ + \frac{1}{\delta}\right\}\right]}$$

$$\overset{(a)}{\leq} \sqrt{\ln\left[\max\left\{e, 9\ln\frac{1}{\epsilon}\ln c\xi[\ln\ln]_+ + \frac{1}{\delta}\right\}\right]}$$

$$\leq \sqrt{3 + [\ln\ln]_+ \frac{1}{\epsilon} + [\ln\ln]_+ c\xi + \ln[\ln\ln]_+ \frac{1}{\delta}}$$

$$\overset{(b)}{\leq} \sqrt{3} + \sqrt{[\ln\ln]_+ c\xi} + \sqrt{[\ln\ln]_+ \frac{1}{\epsilon}} + \sqrt{\ln[\ln\ln]_+ \frac{1}{\delta}}$$

where (a) follows by $a + b + c \leq 3abc$ if $a, b, c \geq 1$, and (b) follows by $\sqrt{\sum_i x_i} \leq \sum_i \sqrt{x_i}$ if $x_i \geq 0$.

Thus, we have

$$\frac{1}{\sqrt{N}}\left(D_0\left(1 + \ln\frac{1}{\delta} + \sqrt{N[\ln\ln]_+ N} + \sqrt{N\ln\frac{1}{\delta}}\right) - N\epsilon\right)$$

$$\leq \sqrt{\ln\frac{1}{\delta}}\left(\frac{2D_0}{\sqrt{\xi}} + D_0\frac{\sqrt{3} + \sqrt{[\ln\ln]_+ c\xi} + \sqrt{[\ln\ln]_+ \frac{1}{\epsilon}} + \sqrt{\ln[\ln\ln]_+ \frac{1}{\delta}}}{\sqrt{\ln\frac{1}{\delta}}} + D_0 - \sqrt{c\xi[\ln\ln]_+ \frac{1}{\epsilon}}\right)$$

$$\overset{(c)}{\leq} \sqrt{\ln\frac{1}{\delta}}\left(\frac{2D_0}{\sqrt{\xi}} + D_0\left(3 + \sqrt{[\ln\ln]_+ c\xi}\right) + D_0 - \sqrt{c\xi}/2\right)$$

$$\overset{(d)}{\leq} -\sqrt{\ln\frac{1}{\delta}/2}$$

(c) follows by $\sqrt{\ln\frac{1}{\delta}} \geq \max\left\{1, \sqrt{\ln[\ln\ln]_+ \frac{1}{\delta}}\right\}$, $D_0 \geq 1$, and $\sqrt{[\ln\ln]_+ \frac{1}{\epsilon}}\left(\frac{D_0}{\sqrt{\ln\frac{1}{\delta}}} - \sqrt{c\xi}\right) \leq D_0 - \sqrt{c\xi} \leq -\sqrt{c\xi}/2$ if $c\xi \geq 4D_0^2$. (d) follows by our choose of $\xi$.

Therefore,

$$\Pr\left(S_N - N\mathbb{E}X_i \leq D_0\left(1 + \ln\frac{1}{\delta} + \sqrt{N[\ln\ln]_+ N} + \sqrt{N\ln\frac{1}{\delta}}\right) - N\epsilon\right)$$

$$\leq \Pr\left(S_N - N\mathbb{E}X_i \leq -\sqrt{N\ln\frac{1}{\delta}/2}\right)$$

which is at most $\delta$ by Hoeffding Bound. $\qquad\square$

**Lemma 3.** *Suppose $\{X_i\}_{i=1}^\infty$ is a sequence of i.i.d. random variables such that $\mathbb{E}X_i \leq 0$, $|X_i| \leq 1$. Let $\delta > 0$. Then with probability at least $1 - \delta$, for all $n$ simultaneously CheckSignificant-Var$(\{X_i\}_{i=1}^n, \delta)$ in Procedure 2 returns false.*

*Proof.* Define $Y_i = X_i - \mathbb{E}X_i$. It is easy to check $\frac{n}{n-1}\left(\sum_{i=1}^n Y_i^2 - \frac{1}{n}\left(\sum_{i=1}^n Y_i\right)^2\right) = \frac{n}{n-1}\left(\sum_{i=1}^n X_i^2 - \frac{1}{n}\left(\sum_{i=1}^n X_i\right)^2\right)$. The result is immediate from Proposition 2. $\qquad\square$

**Lemma 4.** *Suppose $\{X_i\}_{i=1}^{\infty}$ is a sequence of i.i.d. random variables such that $\mathbb{E}X_i > \tau\epsilon$, $|X_i| \leq 1$, $Var(X_i) \leq 2\epsilon$ where $0 < \epsilon \leq 1$, $\tau > 0$. Let $\delta < 1$, $N = \frac{\xi}{\tau\epsilon}\ln\frac{2}{\delta}$ ($\xi$ is a constant specified in the proof). Then with probability at least $1 - \delta$, CheckSignificant-Var$\left(\{X_i\}_{i=1}^{N}, \delta\right)$ in Procedure 2 returns true.*

*Proof.* Let $Y_i = X_i - \mathbb{E}X_i$, $\eta$ be the constant $\eta$ in Lemma 14. Set $\xi = \max(\eta, \frac{16}{\tau} + \frac{8}{3})$.

CheckSignificant-Var$\left(\{X_i\}_{i=1}^{N}, \delta\right)$ returns false if and only if $\sum_{i=1}^{N} X_i \leq q(N, \text{Var}, \delta)$.

By applying Lemma 14 to $X_i$, $\frac{q(N,\text{Var},\delta)}{N} - \mathbb{E}X_i \leq -\tau\epsilon/2$ with probability at least $1 - \delta/2$. Applying Bernstein's inequality to $Y_i$, we have

$$
\begin{aligned}
\Pr\left(\frac{1}{N}\sum_{i=1}^{N} Y_i \leq -\tau\epsilon/2\right) &\leq \exp\left(-\frac{N\left(-\tau\epsilon\right)^2/4}{4\epsilon + 2\tau\epsilon/3}\right) \\
&= \exp\left(-\frac{\xi\ln\frac{2}{\delta}}{16/\tau + 8/3}\right) \\
&\leq \delta/2
\end{aligned}
$$

Thus, by a union bound,

$$
\begin{aligned}
&\Pr\left(\sum_{i=1}^{N} X_i \leq q(N, \text{Var}, \delta)\right) \\
\leq\, &\Pr\left(\frac{q(N, \text{Var}, \delta)}{N} - \mathbb{E}X_i \geq -\tau\epsilon/2\right) \\
&+ \Pr\left(\frac{q(N, \text{Var}, \delta)}{N} - \mathbb{E}X_i \leq -\tau\epsilon/2 \text{ and } \frac{1}{N}\sum_{i=1}^{N} X_i \leq \frac{q(N, \text{Var}, \delta)}{N}\right) \\
\leq\, &\delta/2 + \Pr\left(\frac{q(N, \text{Var}, \delta)}{N} - \mathbb{E}X_i \leq -\tau\epsilon/2 \text{ and } \frac{1}{N}\sum_{i=1}^{N} Y_i \leq \frac{q(n, \text{Var}, \delta)}{N} - \mathbb{E}X_i\right) \\
\leq\, &\delta/2 + \Pr\left(\frac{1}{N}\sum_{i=1}^{N} Y_i \leq -\tau\epsilon/2\right) \\
\leq\, &\delta
\end{aligned}
$$

$\square$

### A.2 The one-dimensional case

*Proof of Theorem 1.* Since $\hat{\theta} = \left(L_{\log\frac{1}{2\epsilon}} + R_{\log\frac{1}{2\epsilon}}\right)/2$ and $R_{\log\frac{1}{2\epsilon}} - L_{\log\frac{1}{2\epsilon}} = 2\epsilon$, $\left|\hat{\theta} - \theta^*\right| > \epsilon$ is equivalent to $\theta^* \notin [L_{\log\frac{1}{2\epsilon}}, R_{\log\frac{1}{2\epsilon}}]$. We have

$$
\begin{aligned}
\Pr\left(\left|\hat{\theta} - \theta^*\right| > \epsilon\right) &= \Pr\left(\theta^* \notin [L_{\log\frac{1}{2\epsilon}}, R_{\log\frac{1}{2\epsilon}}]\right) \\
&= \Pr\left(\exists k : \theta^* \in [L_k, R_k] \text{ and } \theta^* \notin [L_{k+1}, R_{k+1}]\right) \\
&\leq \sum_{k=0}^{\log\frac{1}{2\epsilon}-1} \Pr\left(\theta^* \in [L_k, R_k] \text{ and } \theta^* \notin [L_{k+1}, R_{k+1}]\right)
\end{aligned}
$$

For any $k = 0, \ldots, \log\frac{1}{2\epsilon} - 1$, define $\mathbb{Q}_k = \left\{(p, q) : p, q \in \mathbb{Q} \cap [0, 1] \text{ and } q - p = \left(\frac{3}{4}\right)^k\right\}$ where $\mathbb{Q}$ is the set of rational numbers. Note that $L_k, R_k \in \mathbb{Q}_k$, and $\mathbb{Q}$ is countable. So we have

$$\Pr\left(\theta^* \in [L_k, R_k] \text{ and } \theta^* \notin [L_{k+1}, R_{k+1}]\right)$$

$$= \sum_{(p,q)\in\mathbb{Q}_k: p\leq\theta^*\leq q} \Pr\left(L_k = p, R_k = q \text{ and } \theta^* \notin [L_{k+1}, R_{k+1}]\right)$$

$$= \sum_{(p,q)\in\mathbb{Q}_k: p\leq\theta^*\leq q} \Pr\left(\theta^* \notin [L_{k+1}, R_{k+1}]|L_k = p, R_k = q\right)\Pr\left(L_k = p, R_k = q\right)$$

Define event $E_{k,p,q}$ to be the event $L_k = p, R_k = q$. To show $\Pr\left(\left|\hat{\theta} - \theta^*\right| > \epsilon\right) \leq \frac{\delta}{2}$, it suffices to show $\Pr\left(\theta^* \notin [L_{k+1}, R_{k+1}]|E_{k,p,q}\right) \leq \frac{\delta}{2\log\frac{1}{2\epsilon}}$ for any $k = 0, \ldots, \log\frac{1}{2\epsilon} - 1$, $(p,q) \in \mathbb{Q}_k$ and $p \leq \theta^* \leq q$.

Conditioning on event $E_{k,p,q}$, event $\theta^* \notin [L_{k+1}, R_{k+1}]$ happens only if some calls of CheckSignificant and CheckSignificant-Var between Line 16 and 27 of Algorithm 1 return true incorrectly. In other words, at least one of following events happens for some $n$:

- $O_{k,p,q}^{(1)}$: $\theta^* \in [L_k, U_k]$ and CheckSignificant-Var($\left\{A_i^{(u)} - A_i^{(m)}\right\}_{i=1}^n, \frac{\delta}{4\log\frac{1}{2\epsilon}}$) returns true;

- $O_{k,p,q}^{(2)}$: $\theta^* \in [V_k, R_k]$ and CheckSignificant-Var($\left\{A_i^{(v)} - A_i^{(m)}\right\}_{i=1}^n, \frac{\delta}{4\log\frac{1}{2\epsilon}}$) returns true;

- $O_{k,p,q}^{(3)}$: $\theta^* \in [L_k, U_k]$ and CheckSignificant($\left\{-B_i^{(u)}\right\}_{i=1}^n, \frac{\delta}{4\log\frac{1}{2\epsilon}}$) returns true;

- $O_{k,p,q}^{(4)}$: $\theta^* \in [V_k, R_k]$ and CheckSignificant($\left\{B_i^{(v)}\right\}_{i=1}^n, \frac{\delta}{4\log\frac{1}{2\epsilon}}$) returns true;

Note that since $[U_k, V_k] \subset [L_{k+1}, R_{k+1}]$ for any $k$ by our construction, if $\theta^* \in [U_k, V_k]$ then $\theta^* \in [L_{k+1}, R_{k+1}]$. Besides, event $\theta^* \in [L_k, U_k]$ and event $\theta^* \in [V_k, R_k]$ are mutually exclusive.

Conditioning on event $E_{k,p,q}$, suppose for now $\theta^* \in [L_k, U_k]$.

$$\Pr\left(O_{k,p,q}^{(1)} \mid E_{k,p,q}\right)$$
$$= \Pr\left(\exists n : \text{CheckSignificant-Var}\left(\left\{D_i^{(u,m)}\right\}_{i=1}^n, \frac{\delta}{4\log\frac{1}{2\epsilon}}\right) \text{ returns true} \mid \theta^* \in [L_k, U_k], E_{k,p,q}\right)$$

On event $\theta^* \in [L_k, U_k]$ and $E_{k,p,q}$, the sequences $\left\{A_i^{(u)}\right\}$ and $\left\{A_i^{(m)}\right\}$ are i.i.d., and $\mathbb{E}\left[A_i^{(u)} - A_i^{(m)} \mid \theta^* \in [L_k, U_k], E_{k,p,q}\right] \leq 0$. By Lemma 3, the probability above is at most $\frac{\delta}{4\log\frac{1}{2\epsilon}}$.

Likewise,

$$\Pr\left(O_{k,p,q}^{(3)} \mid E_{k,p,q}\right)$$
$$= \Pr\left(\exists n : \text{CheckSignificant}\left(\left\{-B_i^{(u)}\right\}_{i=1}^n, \frac{\delta}{4\log\frac{1}{2\epsilon}}\right) \text{ returns true} \mid \theta^* \in [L_k, U_k], E_{k,p,q}\right)$$

On event $\theta^* \in [L_k, U_k]$ and $E_{k,p,q}$, the sequence $\left\{B_i^{(u)}\right\}$ is i.i.d., and $\mathbb{E}\left[-B_i^{(u)} \mid \theta^* \in [L_k, U_k], E_{k,p,q}\right] \leq 0$. By Lemma 1, the probability above is at most $\frac{\delta}{4\log\frac{1}{2\epsilon}}$.

Thus, $\Pr\left(\theta^* \notin [L_{k+1}, R_{k+1}] \mid E_{k,p,q}\right) \leq \frac{\delta}{2\log\frac{1}{2\epsilon}}$ when $\theta^* \in [L_k, U_k]$. Similarly, when $\theta^* \in [V_k, R_k]$, we can show $\Pr\left(\theta^* \notin [L_{k+1}, R_{k+1}] \mid E_{k,p,q}\right) \leq \Pr\left(O_{k,p,q}^{(2)} \mid E_{k,p,q}\right) + \Pr\left(O_{k,p,q}^{(4)} \mid E_{k,p,q}\right) \leq \frac{\delta}{2\log\frac{1}{2\epsilon}}$.

Therefore, $\Pr\left(\theta^* \notin [L_{k+1}, R_{k+1}] \mid E_{k,p,q}\right) \leq \frac{\delta}{2\log\frac{1}{2\epsilon}}$, and thus $\Pr\left(\left|\hat{\theta} - \theta^*\right| > \epsilon\right) \leq \delta/2$. $\qquad\square$

*Proof of Theorem 2.* Define $T_k$ to be the number of iterations of the loop at Line 6, $T = \sum_{k=0}^{\log\frac{1}{2\epsilon}-1} T_k$. For any numbers $m_1, m_2, \ldots, m_{\log\frac{1}{2\epsilon}-1}$, we have:

$$
\begin{aligned}
\Pr\left(T \geq m\right) &\leq \Pr\left(\left|\hat{\theta} - \theta^*\right| > \epsilon\right) + \Pr\left(\left|\hat{\theta} - \theta^*\right| < \epsilon \text{ and } T \geq \sum_{k=0}^{\log\frac{1}{2\epsilon}-1} m_k\right) \\
&\leq \frac{\delta}{2} + \Pr\left(T \geq \sum_{k=0}^{\log\frac{1}{2\epsilon}-1} m_k \text{ and } \left|\hat{\theta} - \theta^*\right| < \epsilon\right) \qquad (1) \\
&\leq \frac{\delta}{2} + \sum_{k=0}^{\log\frac{1}{2\epsilon}-1} \Pr\left(T_k \geq m_k \text{ and } \left|\hat{\theta} - \theta^*\right| < \epsilon\right) \\
&\leq \frac{\delta}{2} + \sum_{k=0}^{\log\frac{1}{2\epsilon}-1} \Pr\left(T_k \geq m_k \text{ and } \theta^* \in [L_k, R_k]\right)
\end{aligned}
$$

The first and the third inequality follows by union bounds. The second follows by Theorem 1. The last follows since $\left|\hat{\theta} - \theta^*\right| < \epsilon$ is equivalent to $\theta^* \in [L_{\log\frac{1}{2\epsilon}}, R_{\log\frac{1}{2\epsilon}}]$, which implies $\theta^* \in [L_k, R_k]$ for all $k = 0, \ldots, \log\frac{1}{2\epsilon} - 1$.

We define $\mathbb{Q}_k$ as in the previous proof. For all $k = 0, \ldots, \log\frac{1}{2\epsilon} - 1$,

$$
\begin{aligned}
&\Pr\left(T_k \geq m_k \text{ and } \theta^* \in [L_k, R_k]\right) \\
&= \sum_{(p,q)\in\mathbb{Q}_k : p \leq \theta^* \leq q} \Pr\left(T_k \geq m_k, L_k = p, R_k = q\right) \\
&= \sum_{(p,q)\in\mathbb{Q}_k : p \leq \theta^* \leq q} \Pr\left(T_k \geq m_k | L_k = p, R_k = q\right) \Pr\left(L_k = p, R_k = q\right)
\end{aligned}
$$

Thus, in order to prove the query complexity of Algorithm 1 is $O\left(\sum_{k=0}^{\log\frac{1}{2\epsilon}-1} m_k\right)$, it suffices to show that $\Pr\left(T_k \geq m_k \mid L_k = p, R_k = q\right) \leq \frac{\delta}{2\log\frac{1}{2\epsilon}}$ for any $k = 0, \ldots, \log\frac{1}{2\epsilon} - 1$, $(p,q) \in \mathbb{Q}_k$ and $p \leq \theta^* \leq q$.

For each $k, p, q$, define event $E_{k,p,q}$ to be the event $L_k = p, R_k = q$. Define $l_k = q - p = \left(\frac{3}{4}\right)^k$, $N_k$ to be $\tilde{\Theta}\left(\frac{1}{f(l_k/4)} l_k^{-2\beta}\right)$. The logarithm factor of $N_k$ is to be specified later. Define $S_n^{(u)}$ and $S_n^{(v)}$ to be the size of array $B^{(u)}$ and $B^{(v)}$ before Line 16 respectively.

To show $\Pr\left(T_k \geq N_k \mid E_{k,p,q}\right) \leq \frac{\delta}{2\log\frac{1}{2\epsilon}}$, it suffices to show that on event $E_{k,p,q}$, with probability at least $1 - \frac{\delta}{2\log\frac{1}{2\epsilon}}$, if $n = N_k$ then at least one of the two calls to CheckSignificant between Line 22 and Line 27 will return true.

On event $E_{k,p,q}$, if $\theta^* \in [L_k, M_k]$ (note that on event $E_{k,p,q}$, $L_k$ and $M_k$ are deterministic), then $|V_k - \theta^*| \geq \frac{l_k}{4}$. We will show

$$
p_1 := \Pr\left(\text{CheckSignificant}\left(\left\{B_i^{(v)}\right\}_{i=1}^{S_{N_k}^{(v)}}, \frac{\delta}{4\log\frac{1}{2\epsilon}}\right) \text{ returns false} \mid E_{k,p,q}\right) \leq \frac{\delta}{2\log\frac{1}{2\epsilon}}
$$

To prove this, we will first show that $S_{N_k}^{(v)}$, the length of the array $B^{(v)}$, is large with high probability, and then apply Lemma 2 to show that CheckSignificant will return true if $S_{N_k}^{(v)}$ is large.

By definition, $S_{N_k}^{(v)} = \sum_{i=1}^{N_k} A_i^{(v)}$. By Condition 2, $\mathbb{E}\left[A_i^{(v)} \mid E_{k,p,q}\right] = \Pr\left(Y \neq \perp \mid X = V_k, E_{k,p,q}\right) \geq f\left(\frac{l_k}{4}\right)$.

On event $E_{k,p,q}$, $\left\{A_i^{(v)}\right\}$ is a sequence of i.i.d. random variables. By the multiplicative Chernoff bound, $\Pr\left(S_{N_k}^{(v)} \leq \frac{1}{2}N_k f\left(\frac{l_k}{4}\right) \mid E_{k,p,q}\right) \leq \exp\left(-N_k f\left(\frac{l_k}{4}\right)/8\right)$.

Now,

$$p_1 \leq \Pr\left(\text{CheckSignificant}\left(\left\{B_i^{(v)}\right\}_{i=1}^{S_{N_k}^{(v)}}, \frac{\delta}{4\log\frac{1}{2\epsilon}}\right) \text{ returns false}, S_{N_k}^{(v)} \geq \frac{1}{2}N_k f\left(\frac{l_k}{4}\right) \mid E_{k,p,q}\right)$$

$$+ \Pr\left(S_{N_k}^{(v)} < \frac{1}{2}N_k f\left(\frac{l_k}{4}\right) \mid E_{k,p,q}\right)$$

By Condition 2 and $|V_k - \theta^*| \geq \frac{l_k}{4}$, $\mathbb{E}\left[B_i^{(v)} \mid E_{k,p,q}\right] \geq C\left(\frac{l_k}{4}\right)^\beta$. On event $E_{k,p,q}$, $\left\{B_i^{(v)}\right\}$ is a sequence of i.i.d. random variables. Thus, On event $E_{k,p,q}$, by Lemma 2, with probability at least $1 - \frac{\delta}{4\log\frac{1}{2\epsilon}}$, CheckSignificant will return true if $\frac{1}{2}N_k f\left(\frac{l_k}{4}\right) = \Theta\left(\frac{1}{l_k^{2\beta}}\ln\frac{\ln 1/\epsilon}{\delta}[\ln\ln]_+\frac{1}{l_k^{2\beta}}\right)$. We have already proved $\Pr\left(S_{N_k}^{(v)} \leq \frac{1}{2}N_k f\left(\frac{l_k}{4}\right) \mid E_{k,p,q}\right) \leq \exp\left(-N_k f\left(\frac{l_k}{4}\right)/8\right)$. By setting $N_k = \Theta\left(\frac{1}{f(l_k/4)}l_k^{-2\beta}\ln\frac{\ln 1/\epsilon}{\delta}[\ln\ln]_+\frac{1}{l_k^{2\beta}}\right)$, we can ensure $p_1$ is at most $\delta/2\log\frac{1}{2\epsilon}$.

Now we have proved on event $E_{k,p,q}$, if $\theta^* \in [L_k, M_k]$, then

$$\Pr\left(\text{CheckSignificant}\left(\left\{B_i^{(v)}\right\}_{i=1}^{S_{N_k}^{(v)}}, \frac{\delta}{4\log\frac{1}{2\epsilon}}\right) \text{ returns true} \mid E_{k,p,q}\right) \geq 1 - \frac{\delta}{2\log\frac{1}{2\epsilon}}$$

Likewise, on event $E_{k,p,q}$, if $\theta^* \in [M_k, R_k]$, then

$$\Pr\left(\text{CheckSignificant}\left(\left\{-B_i^{(u)}\right\}_{i=1}^{S_{N_k}^{(u)}}, \frac{\delta}{4\log\frac{1}{2\epsilon}}\right) \text{ returns true} \mid E_{k,p,q}\right) \geq 1 - \frac{\delta}{2\log\frac{1}{2\epsilon}}$$

Therefore, we have shown $\Pr\left(T_k \geq N_k \mid E_{k,p,q}\right) \leq \frac{\delta}{2\log\frac{1}{2\epsilon}}$ for any $k, p, q$. By (1), with probability at least $1 - \delta$, the number of samples queried is at most

$$\sum_{k=0}^{\log\frac{1}{2\epsilon}-1} O\left(\frac{1}{f((\frac{3}{4})^k/4)}\left(\frac{3}{4}\right)^{-2\beta k}\ln\frac{\ln 1/\epsilon}{\delta}[\ln\ln]_+\left(\frac{3}{4}\right)^{-2k\beta}\right)$$

$$= O\left(\frac{\epsilon^{-2\beta}}{f(\epsilon/2)}\ln\frac{1}{\epsilon}\left(\ln\frac{1}{\delta} + \ln\ln\frac{1}{\epsilon}\right)[\ln\ln]_+\frac{1}{\epsilon}\right)$$

$\square$

*Proof of Theorem 3.* For each $k$ in Algorithm 1 at Line 3, Let $l_k = R_k - L_k$. Let $N_k = \eta\frac{1}{f(l_k/4)}\ln\frac{4\log\frac{1}{2\epsilon}}{\delta}$, where $\eta$ is a constant to be specified later. As with the previous proof, it suffices to show $\Pr\left(T_k \geq N_k \mid E_{k,p,q}\right) \leq \frac{\delta}{2\log\frac{1}{2\epsilon}}$ where event $E_{k,p,q}$ is defined to be $L_k = p, R_k = q$, $T_k$ is the number of iterations at the loop at Line 6.

On event $E_{k,p,q}$, we will show that the loop at Line 6 will terminate after $n = N_k$ with probability at least $1 - \frac{\delta}{2\log\frac{1}{2\epsilon}}$.

Suppose for now $\theta^* \in [M_k, R_k]$. Let $Z_i = A_i^{(u)} - A_i^{(m)}$, $\zeta = \theta^* - M_k$. Clearly, $|Z_i| \leq 1$. On event $E_{k,p,q}$, sequence $\{Z_i\}$ is i.i.d.. By Condition 3, $\mathbb{E}[Z_i \mid E_{k,p,q}] = f(\zeta + \frac{l_k}{4}) - f(\zeta) \geq$

$cf(\zeta + \frac{l_k}{4})$ since $\zeta \le \frac{2}{3}(\zeta + \frac{l_k}{4})$. $\mathrm{Var}\left[Z_i | E_{k,p,q}\right] = \mathrm{Var}\left[A_i^{(u)} | E_{k,p,q}\right] + \mathrm{Var}\left[A_i^{(m)} | E_{k,p,q}\right] \overset{(a)}{\le}$
$\mathbb{E}\left[A_i^{(u)} | E_{k,p,q}\right] + \mathbb{E}\left[A_i^{(m)} | E_{k,p,q}\right] = f(\zeta + \frac{l_k}{4}) + f(\zeta) \overset{(b)}{\le} 2f(\zeta + \frac{l_k}{4})$ where (a) follows by $A_i \in \{0,1\}$ and (b) follows by the monotonicity of $f$ . Thus, on event $E_{k,p,q}$, by Lemma 4, if we set $\eta$ sufficiently large (independent of $l_k, \epsilon, \delta$), then with probability at least $1 - \frac{\delta}{4\log\frac{1}{2\epsilon}}$ CheckSignificant-Var$\left(\{Z_i\}_{i=1}^{N_k}, \frac{\delta}{4\log\frac{1}{2\epsilon}}\right)$ in Procedure 2 returns true.

Similarly, we can show that on event $E_{k,p,q}$, if $\theta^* \in [L_k, M_k]$, by Lemma 4, with probability at least $1 - \frac{\delta}{4\log\frac{1}{2\epsilon}}$, CheckSignificant-Var$\left(\left\{A_i^{(v)} - A_i^{(m)}\right\}_{i=1}^{N_k}, \frac{\delta}{4\log\frac{1}{2\epsilon}}\right)$ returns true.

Therefore, the loop at Line 6 will terminate after $n = N_k$ with probability at least $1 - \frac{\delta}{4\log\frac{1}{2\epsilon}}$ on event $E_{k,p,q}$. Therefore, with probability at least $1 - \delta$, the number of samples queried is at most $\sum_{k=0}^{\log\frac{1}{2\epsilon}-1} \frac{1}{f((\frac{3}{4})^k/4)} \ln\frac{\ln 1/\epsilon}{\delta} = O\left(\frac{1}{f(\epsilon/2)} \ln\frac{1}{\epsilon}\left(\ln\frac{1}{\delta} + \ln\ln\frac{1}{\epsilon}\right)\right).$ $\qquad\square$

### A.3 The d-dimensional case

To prove the $d$-dimensional case, we only need to use a union bound to show that with high probability all calls of Algorithm 1 succeed, and consequently the output boundary $g$ produced by polynomial interpolation is close to the true underlying boundary due to the smoothness assumption of $g^*$.

*Proof of Theorem 8.* For $q \in \left\{0, 1, \ldots, \frac{M}{\gamma} - 1\right\}^{d-1}$, define the "polynomial interpolation" version of $g^*$ as

$$g_q^*(\tilde{\boldsymbol{x}}) = \sum_{l \in I_q \cap \mathcal{L}} g^*(l) Q_{q,l}(\tilde{\boldsymbol{x}})$$

Recall that we choose $M = O\left(\epsilon^{-1/\gamma}\right)$.

By Theorem 1, each run of Algorithm 1 at the line 3 of Algorithm 3 will return a $g_l$ such that $\left|g_l - g_q^*(l)\right| \le \epsilon$ with probability at least $1 - \delta/2M^{d-1}$.

$$\|g - g^*\|$$
$$= \sum_{q \in \{0, \ldots, M/\gamma - 1\}^{d-1}} \left\|(g_q - g^*) \mathbb{1}\{\tilde{\boldsymbol{x}} \in I_q\}\right\|$$
$$\le \sum_{q \in \{0, \ldots, M/\gamma - 1\}^{d-1}} \left\|\left(g_q - g_q^*\right) \mathbb{1}\{\tilde{\boldsymbol{x}} \in I_q\}\right\| + \left\|\left(g_q^* - g^*\right) \mathbb{1}\{\tilde{\boldsymbol{x}} \in I_q\}\right\|$$

$$\left\|\left(g_q^* - g^*\right) \mathbb{1}\{\tilde{\boldsymbol{x}} \in I_q\}\right\| = \int_{I_q} \left|g_q^*(\tilde{\boldsymbol{x}}) - g^*(\tilde{\boldsymbol{x}})\right| d\tilde{\boldsymbol{x}}$$
$$= O\left(\int_{I_q} M^{-\gamma} d\tilde{\boldsymbol{x}}\right)$$
$$= O\left(M^{-\gamma - d + 1}\right)$$

The second equality follows from Lemma 3 of [6] that $\left|g_q(\tilde{\boldsymbol{x}}) - g^*(\tilde{\boldsymbol{x}})\right| = O\left(M^{-\gamma}\right)$ since $g^*$ is $\gamma$-Hölder smooth.

$$\left\| (g_q - g_q^*) \, \mathbb{1}\{\tilde{\boldsymbol{x}} \in I_q\} \right\|$$

$$= \sum_{l \in I_q \cap \mathcal{L}} |g_l - g_q^*(l)| \, \|Q_{q,l}\|$$

$$\leq \sum_{l \in I_q \cap \mathcal{L}} \epsilon \, \|Q_q\|$$

$$= O(\epsilon M^{-d+1})$$

Therefore, overall we have $\|g - g^*\| \leq O\left(M^{-\gamma-d+1} + \epsilon M^{-d+1}\right) \left(\frac{M}{\gamma}\right)^{d-1} = O(\epsilon)$. $\qquad\square$

*Proof of Theorem 9.* By Theorem 2, each run of Algorithm 1 at the line 3 of Algorithm 3 will make $\tilde{O}\left(\frac{d}{f(\epsilon/2)}\epsilon^{-2\beta}\right)$ queries with probability at least $1 - \delta/M^{d-1}$, thus by a union bound, the total number of queries made is $\tilde{O}\left(\frac{d}{f(\epsilon/2)}\epsilon^{-2\beta-\frac{d-1}{\gamma}}\right)$ with probability at least $1 - \delta$. $\qquad\square$

*Proof of Theorem 10.* The proof is similar to the previous proof. $\qquad\square$

## B  Proof of lower bounds

First, we introduce some notations for this section. Given a labeler $L$ and an active learning algorithm $\mathcal{A}$, denote by $P_{L,\mathcal{A}}^n$ the distribution of $n$ samples $\{(X_i, Y_i)\}_{i=1}^n$ where $Y_i$ is drawn from distribution $P_L(Y|X_i)$ and $X_i$ is drawn by the active learning algorithm based solely on the knowledge of $\{(X_j, Y_j)\}_{j=1}^{i-1}$. We will drop the subscripts from $P_{L,\mathcal{A}}^n$ and $P_{\mathcal{L}}(Y|X)$ when it is clear from the context. For a sequence $\{X_i\}_{i=1}^\infty$ denote by $X^n$ the subsequence $\{X_1, \ldots, X_n\}$.

**Definition 1.** For any distributions $P, Q$ on a countable support, define KL-divergence as $d_{\mathrm{KL}}(P, Q) = \sum_x P(x) \ln \frac{P(x)}{Q(x)}$. For two random variables $X, Y$, define the mutual information as $I(X; Y) = d_{\mathrm{KL}}(P(X,Y) \| P(X)P(Y))$.

We will use Fano's method shown as below to prove the lower bounds.

**Lemma 5.** *Let $\Theta$ be a class of parameters, and $\{P_\theta : \theta \in \Theta\}$ be a class of probability distributions indexed by $\Theta$ over some sample space $\mathcal{X}$. Let $d : \Theta \times \Theta \to \mathbb{R}$ be a semi-metric. Let $\mathcal{V} = \{\theta_1, \ldots, \theta_M\} \subseteq \Theta$ such that $\forall i \neq j, d(\theta_i, \theta_j) \geq 2s > 0$. Let $\bar{P} = \frac{1}{M}\sum_{\theta \in \mathcal{V}} P_\theta$. If $d_{KL}(P_\theta \| \bar{P}) \leq \delta$ for any $\theta \in \mathcal{V}$, then for any algorithm $\hat{\theta}$ that given a sample $X$ drawn from $P_\theta$ outputs $\hat{\theta}(X) \in \Theta$, the following inequality holds:*

$$\sup_{\theta \in \Theta} P_\theta\left(d(\theta, \hat{\theta}(X)) \geq s\right) \geq 1 - \frac{\delta + \ln 2}{\ln M}$$

*Proof.* For any algorithm $\hat{\theta}$, define a test function $\hat{\Psi} : \mathcal{X} \to \{1, \ldots, M\}$ such that $\hat{\Psi}(X) = \arg\min_{i \in \{1,\ldots,M\}} d(\hat{\theta}(X), \theta_i)$. We have

$$\sup_{\theta \in \Theta} P_\theta\left(d(\theta, \hat{\theta}(X)) \geq s\right) \geq \max_{\theta \in \mathcal{V}} P_\theta\left(d(\theta, \hat{\theta}(X)) \geq s\right) \geq \max_{i \in \{1,\ldots,M\}} P_{\theta_i}\left(\hat{\Psi}(X) \neq i\right)$$

Let $V$ be a random variable uniformly taking values from $\mathcal{V}$, and $X$ be drawn from $P_V$. By Fano's Inequality, for any test function $\Psi : \mathcal{X} \to \{1, \ldots, M\}$

$$\max_{i \in \{1,\ldots,M\}} P_{\theta_i}(\Psi(X) \neq i) \geq 1 - \frac{I(V; X) + \ln 2}{\ln M}$$

The desired result follows by the fact that $I(V; X) = \frac{1}{M}\sum_{\theta \in \mathcal{V}} d_{\mathrm{KL}}(P_\theta \| \bar{P})$. $\qquad\square$

## B.1 The one dimensional case

*Proof of Theorem 5.* [2] Without lose of generality, let $C = C' = 1$ ($C$ is defined in Condition 2). Let $\epsilon \le \frac{1}{4} \min \left\{ \left(\frac{1}{2}\right)^{1/\beta}, \left(\frac{4}{5}\right)^{1/\alpha}, \frac{1}{4} \right\}$. We will prove the desired result using Lemma 5.

First, we construct $\mathcal{V}$ and $P_\theta$. For any $k \in \{0, 1, 2, 3\}$, let $P_{L_k}(Y \mid X)$ be the distribution of the labeler $L_k$'s response with the ground truth $\theta_k = k\epsilon$:

$$
P_{L_k}(Y = \perp \mid x) = 1 - \left| x - \frac{1}{2} - k\epsilon \right|^\alpha
$$

$$
P_{L_k}(Y = 0 \mid x) = 
\begin{cases}
\left( x - \frac{1}{2} - k\epsilon \right)^\alpha \left( 1 - \left( x - \frac{1}{2} - k\epsilon \right)^\beta \right) / 2 & x > \frac{1}{2} + k\epsilon \\
\left( \frac{1}{2} + k\epsilon - x \right)^\alpha \left( 1 + \left( \frac{1}{2} + k\epsilon - x \right)^\beta \right) / 2 & x \le \frac{1}{2} + k\epsilon
\end{cases}
$$

$$
P_{L_k}(Y = 1 \mid x) = 
\begin{cases}
\left( x - \frac{1}{2} - k\epsilon \right)^\alpha \left( 1 + \left( x - \frac{1}{2} - k\epsilon \right)^\beta \right) / 2 & x > \frac{1}{2} + k\epsilon \\
\left( \frac{1}{2} + k\epsilon - x \right)^\alpha \left( 1 - \left( \frac{1}{2} + k\epsilon - x \right)^\beta \right) / 2 & x \le \frac{1}{2} + k\epsilon
\end{cases}
$$

Clearly, $P_{L_k}$ complies with Conditions 1, 2 and 3.

Define $P_k^n$ to be the distribution of $n$ samples $\{(X_i, Y_i)\}_{i=1}^n$ where $Y_i$ is drawn from distribution $P_{L_k}(Y|X_i)$ and $X_i$ is drawn by the active learning algorithm based solely on the knowledge of $\{(X_j, Y_j)\}_{j=1}^{i-1}$.

Define $\bar{P}_L = \frac{1}{4} \sum_j P_{L_j}$ and $\bar{P}^n = \frac{1}{4} \sum_j P_k^n$. We take $\Theta$ to be $[0, 1]$, and $d(\theta_1, \theta_2) = |\theta_1 - \theta_2|$ in Lemma 5. To use Lemma 5, we need to bound $d_{\mathrm{KL}}\left( P_k^n \parallel \bar{P}^n \right)$ for $k \in \{0, 1, 2, 3\}$.

For any $k \in \{0, 1, 2, 3\}$,

$$
\begin{aligned}
& d_{\mathrm{KL}}\left( P_k^n \parallel \bar{P}_0^n \right) \\
=& \mathbb{E}_{P_k^n} \left( \ln \frac{P_k^n \left( \{(X_i, Y_i)\}_{i=1}^n \right)}{\bar{P}^n \left( \{(X_i, Y_i)\}_{i=1}^n \right)} \right) \\
=& \mathbb{E}_{P_k^n} \left( \ln \frac{P_k^n (X_1) P_k^n (Y_1 \mid X_1) P_k^n (X_2 \mid X_1, Y_1) \cdots P_k^n (Y_n \mid X_1, Y_1, \ldots, X_n)}{\bar{P}^n (X_1) \bar{P}^n (Y_1 \mid X_1) \bar{P}^n (X_2 \mid X_1, Y_1) \cdots \bar{P}^n (Y_n \mid X_1, Y_1, \ldots, X_n)} \right) \\
\overset{(a)}{=}& \mathbb{E}_{P_k^n} \left( \ln \frac{\Pi_{i=1}^n P_{L_k}(Y_i|X_i)}{\Pi_{i=1}^n \bar{P}_L(Y_i|X_i)} \right) \\
=& \sum_{i=1}^n \mathbb{E}_{P_k^n} \left( \mathbb{E}_{P_k^n} \left( \ln \frac{P_{L_k}(Y_i|X_i)}{\bar{P}_L(Y_i|X_i)} \mid X^n \right) \right) \\
\le& n \max_{x \in [0,1]} d_{\mathrm{KL}}\left( P_{L_k}(Y \mid x) \parallel \bar{P}_L(Y \mid x) \right)
\end{aligned}
\tag{2}
$$

(a) follows by the fact that $P_k^n (X_{i+1} \mid X_1, Y_1, \ldots X_i, Y_i) = \bar{P}^n (X_{i+1} \mid X_1, Y_1, \ldots, X_i, Y_i)$ since $X_{i+1}$ is drawn by the same active learning algorithm based solely on the knowledge of $\{(X_j, Y_j)\}_{j=1}^i$ regardless of the labeler's response distribution, and the fact that $P_k^n (Y_i \mid X_1, Y_1, \ldots, X_i) = P_{\mathcal{L}_k}(Y_i|X_i)$ and $\bar{P}^n (Y_i \mid X_1, Y_1, \ldots, X_i) = \bar{P}_L(Y_i|X_i)$ by definition.

For any $k \in \{1, 2, 3\}, x \in [0, 1]$,

$$
\bar{P}_L(\cdot \mid x) \ge \frac{P_{L_0}(\cdot \mid x) + P_{L_k}(\cdot \mid x)}{4}
\tag{3}
$$

For any $k \in \{0, 1, 2, 3\}, x \in [0, 1], y \in \{1, -1, \perp\}$

$$\left(\bar{P}_L(Y=y\mid x) - P_{L_k}(Y=y\mid x)\right)^2$$

$$= \left(\sum_j \frac{1}{4}\left(P_{L_j}(Y=y\mid x) - P_{L_0}(Y=y\mid x)\right) + \left(P_{L_0}(Y=y\mid \boldsymbol{x}) - P_{L_k}(Y=y\mid x)\right)\right)^2$$

$$\leq \left(\frac{5}{16}\sum_{j>0}\left(P_{L_j}(Y=y\mid x) - P_{L_0}(Y=y\mid x)\right)^2 + 5\left(P_{L_0}(Y=y\mid x) - P_{L_k}(Y=y\mid x)\right)^2\right)$$

$$\leq 6\sum_{j>0}\left(P_{L_j}(Y=y\mid x) - P_{L_0}(Y=y\mid x)\right)^2 \tag{4}$$

where the first inequality follows by $\left(\sum_{i=0}^4 a_i\right)^2 \leq 5\sum_{i=0}^4 a_i^2$ by letting $a_j = \frac{1}{4}\left(P_{L_j}(Y=y\mid x) - P_{L_0}(Y=y\mid x)\right)$ for $j=0,\dots,3$ and $a_4 = P_{L_0}(Y=y\mid \boldsymbol{x}) - P_{L_k}(Y=y\mid x)$, and noting that $a_0 = 0$ under this setting.

Thus,

$$d_{\mathrm{KL}}\left(P_{L_k}(Y\mid x)\,\|\,\bar{P}_L(Y\mid x)\right)$$

$$\leq \sum_y \frac{1}{\bar{P}_L(Y=y\mid \boldsymbol{x})}\left(P_{L_k}(Y=y\mid x) - \bar{P}_L(Y=y\mid x)\right)^2$$

$$\leq 24\sum_{j>0}\sum_y \frac{1}{P_{L_j}(y\mid x) + P_{L_0}(y\mid x)}\left(P_{L_j}(Y=y\mid x) - P_{L_0}(Y=y\mid x)\right)^2$$

$$\leq O(\epsilon^\alpha)$$

The first inequality follows from Lemma 10. The second inequality follows by (3) and (4). The last inequality follows by applying Lemma 11 to $P_{L_0}(\cdot\mid x)$ and $P_{L_j}(\cdot\mid x)$ and the assumption $\alpha \leq 2$.

Therefore, we have $d_{\mathrm{KL}}\left(P_k^n\,\|\,\bar{P}_0^n\right) = nO(\epsilon^\alpha)$. By setting $n = \epsilon^{-\alpha}$, we get $d_{\mathrm{KL}}\left(P_k^n\,\|\,\bar{P}_0^n\right) \leq O(1)$, and thus by Lemma 5,

$$\sup_\theta P_\theta\left(d(\theta, \hat{\theta}(X)) \geq \Omega(\epsilon)\right) \geq 1 - \frac{O(1) + \ln 2}{\ln 4} = O(1)$$

$\square$

## B.2  The d-dimensional case

Again, we will use Lemma 5 to prove the lower bounds for $d$-dimensional cases. We first construct $\{P_\theta : \theta \in \Theta\}$ using a similar idea with [6], and then use Lemma 12 to select a subset $\tilde{\Theta} \subset \Theta$ to apply Lemma 5.

*Proof of Theorem 6.*  Again, without lose of generality, let $C = 1$. Recall that for $\boldsymbol{x} = (x_1, \dots, x_d) \in \mathbb{R}^d$, we have defined $\tilde{\boldsymbol{x}}$ to be $(x_1, \dots, x_{d-1})$. Define $m = \left(\frac{1}{\epsilon}\right)^{1/\gamma}$. $\mathcal{L} = \left\{0, \frac{1}{m}, \dots, \frac{m-1}{m}\right\}^{d-1}$, $h(\tilde{\boldsymbol{x}}) = \Pi_{i=1}^{d-1}\exp\left(-\frac{1}{1-4x_i^2}\right)\mathbb{1}\left\{|x_i| < \frac{1}{2}\right\}$, $\phi_l(\tilde{\boldsymbol{x}}) = Km^{-\gamma}h(m(\tilde{\boldsymbol{x}}-l)-\frac{1}{2})$ where $l \in \mathcal{L}$. It is easy to check $\phi_l(\tilde{\boldsymbol{x}})$ is $(K,\gamma)$-Hölder smooth and has bounded support $[l_1, l_1+\frac{1}{m}]\times\cdots\times[l_{d-1}, l_{d-1}+\frac{1}{m}]$, which implies that for different $l_1, l_2 \in \mathcal{L}$, the support of $\phi_{l_1}$ and $\phi_{l_2}$ do not intersect.

Let $\Omega = \{0,1\}^{m^{d-1}}$. For any $\boldsymbol{\omega} \in \Omega$, define $g_{\boldsymbol{\omega}}(\tilde{\boldsymbol{x}}) = \sum_{l \in \mathcal{L}} \omega_l \phi_l(\tilde{\boldsymbol{x}})$. For each $\boldsymbol{\omega} \in \Omega$, define the conditional distribution of labeler $L_{\boldsymbol{\omega}}$'s response as follows:

For $x_d \leq A$, $P_{L_{\boldsymbol{\omega}}}(y = \perp \mid \boldsymbol{x}) = 1 - f(A)$, $P_{L_{\boldsymbol{\omega}}}(y \neq \mathbb{I}(x_d > g_{\boldsymbol{\omega}}(\tilde{\boldsymbol{x}}))\mid \boldsymbol{x}, y \neq \perp) = \frac{1}{2}\left(1 - |x_d - g_{\boldsymbol{\omega}}(\tilde{\boldsymbol{x}})|^\beta\right)$;

For $x_d \geq A$, $P_{L_{\boldsymbol{\omega}}}(y =\perp |\boldsymbol{x}) = 1 - f(x_d)$, $P_{L_{\boldsymbol{\omega}}}(y \neq \mathbb{I}(x_d > g_{\omega}(\tilde{\boldsymbol{x}}))|\boldsymbol{x}, y \neq\perp) = \frac{1}{2}\left(1 - x_d^{\beta}\right)$.

Here, $A = c \max \phi(\tilde{\boldsymbol{x}}) = c'\epsilon$ for some constants $c, c'$.

It can be easily verified that $P_{L_{\boldsymbol{\omega}}}$ satisfies Conditions 1 and 2. Note that $g_{\boldsymbol{\omega}}(\tilde{\boldsymbol{x}})$ can be seen as the underlying decision boundary for labeler $P_{L_{\boldsymbol{\omega}}}$.

Define $P_{\boldsymbol{\omega}}^n$ to be the distribution of $n$ samples $\{(X_i, Y_i)\}_{i=1}^n$ where $Y_i$ is drawn from distribution $P_{L_{\boldsymbol{\omega}}}(Y|X_i)$ and $X_i$ is drawn by the active learning algorithm based solely on the knowledge of $\{(X_j, Y_j)\}_{j=1}^{i-1}$.

By Lemma 12, when $\epsilon$ is small enough so that $m^{d-1}$ is large enough, there is a subset $\{\boldsymbol{\omega}^{(1)}, \ldots, \boldsymbol{\omega}^{(M)}\} \subset \Omega$ such that $\left\|\boldsymbol{\omega}^{(i)} - \boldsymbol{\omega}^{(j)}\right\|_0 \geq m^{d-1}/12$ for any $0 \leq i < j \leq M$ and $M \geq 2^{m^{d-1}/48}$. Define $P_i^n = P_{\boldsymbol{\omega}^{(i)}}^n$, $\bar{P}^n = \frac{1}{M}\sum_{i=1}^M P_i^n$.

Next, we will apply Lemma 5 to $\{\boldsymbol{\omega}^{(1)}, \ldots, \boldsymbol{\omega}^{(M)}\}$ with $d(\boldsymbol{\omega}^{(i)}, \boldsymbol{\omega}^{(j)}) = \|g_{\boldsymbol{\omega}^{(i)}} - g_{\boldsymbol{\omega}^{(j)}}\|$. We will lower-bound $d(\boldsymbol{\omega}^{(i)}, \boldsymbol{\omega}^{(j)})$ and upper-bound $d_{\text{KL}}\left(P_i^n \parallel \bar{P}^n\right)$.

For any $1 \leq i < j \leq M$,

$$
\begin{aligned}
&\|g_{\boldsymbol{\omega}^{(i)}} - g_{\boldsymbol{\omega}^{(j)}}\| \\
&= \sum_{l \in \{1,\ldots,m\}^{d-1}} \left|\omega_l^{(i)} - \omega_l^{(j)}\right| Km^{-\gamma-(d-1)} \|h\| \\
&\geq m^{d-1}/12 * Km^{-\gamma-(d-1)} \|h\| \\
&= Km^{-\gamma} \|h\| /12 \\
&= \Theta\left(\epsilon\right)
\end{aligned}
$$

By the convexity of KL-divergence, $d_{\text{KL}}\left(P_i^n \parallel \bar{P}^n\right) \leq \frac{1}{M}\sum_{j=1}^M d_{\text{KL}}\left(P_i^n \parallel P_j^n\right)$, so it suffices to upper-bound $d_{\text{KL}}\left(P_i^n \parallel P_j^n\right)$ for any $i, j$.

For any $1 < i, j \leq M$,

$$
\begin{aligned}
&d_{\text{KL}}\left(P_i^n \parallel P_j^n\right) \\
&\leq n \max_{\boldsymbol{x} \in [0,1]^d} d_{\text{KL}}\left(P_{L_{\boldsymbol{\omega}^{(i)}}}^n(Y \mid \boldsymbol{x}) \parallel P_{L_{\boldsymbol{\omega}^{(j)}}}^n(Y \mid \boldsymbol{x})\right) \\
&= n \max_{\boldsymbol{x} \in [0,1]^d} P_{L_{\boldsymbol{\omega}^{(i)}}}^n(Y \neq\perp| \boldsymbol{x}) d_{\text{KL}}\left(P_{L_{\boldsymbol{\omega}^{(i)}}}^n(Y \mid \boldsymbol{x}, Y \neq\perp) \parallel P_{L_{\boldsymbol{\omega}^{(j)}}}^n(Y \mid \boldsymbol{x}, Y \neq\perp)\right)
\end{aligned}
$$

The inequality follows as (2) in the proof of Theorem 5. The equality follows since $P_{\boldsymbol{\omega}}(y =\perp |\boldsymbol{x})$ is the same for all $\boldsymbol{\omega} \in \Omega$.

If $x_d \geq A$, then $P_{L_{\boldsymbol{\omega}^{(i)}}}^n(Y \mid \boldsymbol{x}, Y \neq\perp) = P_{L_{\boldsymbol{\omega}^{(j)}}}^n(Y \mid \boldsymbol{x}, Y \neq\perp)$, so $d_{\text{KL}}\left(P_{L_{\boldsymbol{\omega}^{(i)}}}^n(Y \mid \boldsymbol{x}, Y \neq\perp) \parallel P_{L_{\boldsymbol{\omega}^{(j)}}}^n(Y \mid \boldsymbol{x}, Y \neq\perp)\right) = 0$. If $x_d < A$, then $P_{L_{\boldsymbol{\omega}^{(i)}}}^n(Y \neq\perp| \boldsymbol{x}) = f(A)$. Therefore,

$$
d_{\text{KL}}\left(P_i^n \parallel P_j^n\right) \leq nf(A) \max_{\boldsymbol{x} \in [0,1]^d} d_{\text{KL}}\left(P_{L_{\boldsymbol{\omega}^{(i)}}}^n(Y \mid \boldsymbol{x}, Y \neq\perp) \parallel P_{L_{\boldsymbol{\omega}^{(j)}}}^n(Y \mid \boldsymbol{x}, Y \neq\perp)\right)
$$

.

Apply Lemma 10 to $P_{L_{\boldsymbol{\omega}^{(i)}}}^n(Y \mid \boldsymbol{x}, Y \neq\perp)$ and $P_{L_{\boldsymbol{\omega}^{(i)}}}^n(Y \mid \boldsymbol{x}, Y \neq\perp)$, and noting they are bounded above by a constant, we have $\max_{\boldsymbol{x} \in [0,1]^d} d_{\text{KL}}\left(P_{L_{\boldsymbol{\omega}^{(i)}}}^n(Y \mid \boldsymbol{x}, Y \neq\perp) \parallel P_{L_{\boldsymbol{\omega}^{(j)}}}^n(Y \mid \boldsymbol{x}, Y \neq\perp)\right) = O\left(A^{2\beta}\right)$. Thus,

$$
d_{\text{KL}}\left(P_i^n \parallel P_j^n\right) \leq nf(A)O\left(A^{2\beta}\right) = nf(c'\epsilon)O(\epsilon^{2\beta})
$$

By setting $n = \frac{1}{f(c'\epsilon)}\epsilon^{-2\beta-\frac{d-1}{\gamma}}$, we get $d_{\text{KL}}\left(P_i^n \parallel P_j^n\right) \leq O\left(\epsilon^{-\frac{d-1}{\gamma}}\right)$. The desired results follows by Lemma 5. $\qquad\square$

The proof of Theorem 7 follows the same structure.

*Proof of Theorem 7.* As in the proof of Theorem 6, let $C = C' = 1$, and define $m = \left(\frac{1}{\epsilon}\right)^{1/\gamma}$. $\mathcal{L} = \left\{0, \frac{1}{m}, \dots, \frac{m-1}{m}\right\}^{d-1}$, $h(\tilde{\boldsymbol{x}}) = \Pi_{i=1}^{d-1} \exp\left(-\frac{1}{1-4x_i^2}\right) \mathbb{1}\left\{|x_i| < \frac{1}{2}\right\}$, $\phi_l(\tilde{\boldsymbol{x}}) = Km^{-\gamma}h(m(\tilde{\boldsymbol{x}} - l) - \frac{1}{2})$ where $l \in \mathcal{L}$. Let $\Omega = \{0,1\}^{m^{d-1}}$. For any $\boldsymbol{\omega} \in \Omega$, define $g_{\boldsymbol{\omega}}(\tilde{\boldsymbol{x}}) = \frac{1}{2} + \sum_{l \in \mathcal{L}} \omega_l \phi_l(\tilde{\boldsymbol{x}})$, which can be seen as a decision boundary. $A = \max \phi(\tilde{\boldsymbol{x}}) = c'\epsilon$ for some constants $c'$.

Let $g_+(\tilde{\boldsymbol{x}}) = g_{(1,1,\dots,1)}(\tilde{\boldsymbol{x}}) = \sum_{l \in \mathcal{L}} \phi_l(\tilde{\boldsymbol{x}})$, $g_-(\tilde{\boldsymbol{x}}) = g_{(0,0,\dots,0)}(\tilde{\boldsymbol{x}}) = 0$. In other words, $g_+$ is the "highest" boundary, and $g_-$ is the "lowest" boundary.

For each $\boldsymbol{\omega} \in \Omega$, define the conditional distribution of labeler $L_{\boldsymbol{\omega}}$'s response as follows:

$$P_{L_{\boldsymbol{\omega}}}(y = \perp | \boldsymbol{x}) = 1 - |x_d - g_{\boldsymbol{\omega}}(\tilde{\boldsymbol{x}})|^\alpha$$

$$P_{L_{\boldsymbol{\omega}}}(y \neq \mathbb{I}(x_d > g_{\boldsymbol{\omega}}(\tilde{\boldsymbol{x}}))|\boldsymbol{x}, y \neq \perp) = \frac{1}{2}\left(1 - |x_d - g_{\boldsymbol{\omega}}(\tilde{\boldsymbol{x}})|^\beta\right)$$

It can be easily verified that $P_{L_{\boldsymbol{\omega}}}$ satisfies Conditions 1, 2, and 3.

Let $P_+(\cdot | \boldsymbol{x}) = P_{L_{(1,1,\dots,1)}}(\cdot | \boldsymbol{x})$, $P_-(\cdot | \boldsymbol{x}) = P_{L_{(0,0,\dots,0)}}(\cdot | \boldsymbol{x})$. By the construction of $g$, for any $\boldsymbol{x} \in [0,1]^d$, any $\boldsymbol{\omega} \in \Omega$, $P_{L_{\boldsymbol{\omega}}}(\cdot | \boldsymbol{x})$ equals either $P_+(\cdot | \boldsymbol{x})$ or $P_-(\cdot | \boldsymbol{x})$.

Define $P_{\boldsymbol{\omega}}^n$ to be the distribution of $n$ samples $\{(X_i, Y_i)\}_{i=1}^n$ where $Y_i$ is drawn from distribution $P_{L_{\boldsymbol{\omega}}}(Y|X_i)$ and $X_i$ is drawn by the active learning algorithm based solely on the knowledge of $\{(X_j, Y_j)\}_{j=1}^{i-1}$.

By Lemma 12, when $\epsilon$ is small enough so that $m^{d-1}$ is large enough,, there is a subset $\Omega' = \left\{\boldsymbol{\omega}^{(1)}, \dots, \boldsymbol{\omega}^{(M)}\right\} \subset \Omega$ such that (i) (well-separated) $\left\|\boldsymbol{\omega}^{(i)} - \boldsymbol{\omega}^{(j)}\right\|_0 \geq m^{d-1}/12$ for any $0 \leq i < j \leq M$, $M \geq 2^{m^{d-1}/48}$; and (ii) (well-balanced) for any $j = 1, \dots, m^{d-1}$, $\frac{1}{24} \leq \frac{1}{M}\sum_{i=1}^M \boldsymbol{\omega}_j^{(i)} \leq \frac{3}{24}$.

Define $P_i^n = P_{\boldsymbol{\omega}^{(i)}}^n$, $\bar{P}^n = \frac{1}{M}\sum_{i=1}^M P_i^n$. Define $P_{L_i} = P_{L_{\boldsymbol{\omega}^{(i)}}}$, $\bar{P}_L = \frac{1}{M}\sum_{i=1}^M P_{L_i}$. By the well-balanced property, for any $\boldsymbol{x} \in [0,1]^d$, $\bar{P}_L(\cdot | \boldsymbol{x})$ is between $\frac{1}{24}P_+(\cdot | \boldsymbol{x}) + \frac{23}{24}P_-(\cdot | \boldsymbol{x})$ and $\frac{3}{24}P_+(\cdot | \boldsymbol{x}) + \frac{21}{24}P_-(\cdot | \boldsymbol{x})$. Therefore

$$\bar{P}_L(\cdot | \boldsymbol{x}) \geq \frac{1}{24}\left(P_+(\cdot | \boldsymbol{x}) + P_-(\cdot | \boldsymbol{x})\right) \tag{5}$$

Moreover, since $P_{L_i}(\cdot | \boldsymbol{x})$ can only take $P_+(\cdot | \boldsymbol{x})$ or $P_-(\cdot | \boldsymbol{x})$ for any $\boldsymbol{x}$,

$$\left|P_{L_i}(\cdot | \boldsymbol{x}) - \bar{P}_{\mathcal{L}}(\cdot | \boldsymbol{x})\right| \leq |P_+(\cdot | \boldsymbol{x}) - P_-(\cdot | \boldsymbol{x})| \tag{6}$$

Next, we will apply Lemma 5 to $\left\{\boldsymbol{\omega}^{(1)}, \dots, \boldsymbol{\omega}^{(M)}\right\}$ with $d(\boldsymbol{\omega}^{(i)}, \boldsymbol{\omega}^{(j)}) = \|g_{\boldsymbol{\omega}^{(i)}} - g_{\boldsymbol{\omega}^{(j)}}\|$. We already know from the proof of Theorem 6 $\|g_{\boldsymbol{\omega}^{(i)}} - g_{\boldsymbol{\omega}^{(j)}}\| = \Omega(\epsilon)$.

For any $0 < i \leq M$, $d_{\mathrm{KL}}\left(P_i^n \| \bar{P}_0^n\right) \leq n \max_{\boldsymbol{x} \in [0,1]^d} d_{\mathrm{KL}}\left(P_{L_i}(Y | \boldsymbol{x}) \| \bar{P}_L(Y | \boldsymbol{x})\right)$. For any $\boldsymbol{x} \in [0,1]^d$,

$$d_{\mathrm{KL}}\left(P_{L_i}(Y | \boldsymbol{x}) \| \bar{P}_L(Y | \boldsymbol{x})\right)$$
$$\leq \sum_y \frac{1}{\bar{P}_L(Y = y | \boldsymbol{x})}\left(P_{L_i}(Y = y | \boldsymbol{x}) - \bar{P}_L(Y = y | \boldsymbol{x})\right)^2$$
$$\leq \sum_y \frac{24}{P_+(y | \boldsymbol{x}) + P_-(y | \boldsymbol{x})}\left(P_+(Y = y | \boldsymbol{x}) - P_-(Y = y | \boldsymbol{x})\right)^2$$
$$\leq O(A^\alpha)$$

The first inequality follows from Lemma 10. The second inequality follows by (5) and (6). The last inequality follows by applying Lemma 11 to $P_+(\cdot | \boldsymbol{x})$ and $P_-(\cdot | \boldsymbol{x})$, setting the $\epsilon$ in Lemma 11 to be $g_{\boldsymbol{\omega}}(\tilde{\boldsymbol{x}})$, and using $g_{\boldsymbol{\omega}}(\tilde{\boldsymbol{x}}) \leq A$ and the assumption $\alpha \leq 2$.

Therefore, we have

$$d_{\mathrm{KL}}\left(P_i^n \parallel P_0^n\right) \leq nO\left(A^\alpha\right) = nO(\epsilon^\alpha)$$

By setting $n = \epsilon^{-\alpha - \frac{d-1}{\gamma}}$, we get $d_{\mathrm{KL}}\left(P_i^n \parallel P_0^n\right) \leq O\left(\epsilon^{-\frac{d-1}{\gamma}}\right)$. Thus by Lemma 5,

$$\sup_\theta P_\theta\left(d(\theta, \hat{\theta}(X)) \geq \Omega\left(\epsilon\right)\right) \geq 1 - \frac{O\left(\epsilon^{-\frac{d-1}{\gamma}}\right) + \ln 2}{\epsilon^{-\frac{d-1}{\gamma}}/48} = O\left(1\right)$$

, from which the desired result follows. $\qquad\square$

## C   Technical lemmas

### C.1   Concentration bounds

In this subsection, we define $Y_1, Y_2, \ldots$ to be a sequence of i.i.d. random variables. Assume $Y_1 \in [-2, 2]$, $\mathbb{E}Y_1 = 0$, $\mathrm{Var}(Y_1) = \sigma^2 \leq 4$. Define $V_n = \frac{n}{n-1}\left(\sum_{i=1}^n Y_i^2 - \frac{1}{n}\left(\sum_{i=1}^n Y_i\right)^2\right)$. It is easy to check $\mathbb{E}V_n = n\sigma^2$.

We need following two results from [21]

**Lemma 6.** *([21], Theorem 2) Take any $0 < \delta < 1$. Then there is an absolute constant $D_0$ such that with probability at least $1 - \delta$, for all $n$ simultaneously,*

$$\left|\sum_{i=1}^n Y_i\right| \leq D_0\left(1 + \ln\frac{1}{\delta} + \sqrt{n\sigma^2\left[\ln\ln\right]_+\left(n\sigma^2\right) + n\sigma^2\ln\frac{1}{\delta}}\right)$$

**Lemma 7.** *([21], Lemma 3) Take any $0 < \delta < 1$. Then there is an absolute constant $K_0$ such that with probability at least $1 - \delta$, for all $n$ simultaneously,*

$$n\sigma^2 \leq K_0\left(1 + \ln\frac{1}{\delta} + \sum_{i=1}^n Y_i^2\right)$$

We note that Proposition 1 is immediate from Lemma 6 since $\mathrm{Var}(Y_i) \leq 4$.

**Lemma 8.** *Take any $0 < \delta < 1$. Then there is an absolute constant $K_3$ such that with probability at least $1 - \delta$, for all $n \geq \ln\frac{1}{\delta}$ simultaneously,*

$$n\sigma^2 \leq K_3\left(1 + \ln\frac{1}{\delta} + V_n\right)$$

*Proof.* By Lemma 7, with probability at least $1 - \delta/2$, for all $n$,

$$n\sigma^2 \leq K_0\left(\sum_{i=1}^n Y_i^2 + \ln\frac{2}{\delta} + 1\right) = K_0\left(\frac{n-1}{n}V_n + \frac{1}{n}\left(\sum_{i=1}^n Y_i\right)^2 + \ln\frac{2}{\delta} + 1\right)$$

By Lemma 6, with probability at least $1 - \delta/2$, for all $n$,

$$\frac{1}{n}\left(\sum_{i=1}^{n} Y_i\right)^2 < \frac{1}{n}\left(D_0\left(1 + \ln\frac{2}{\delta} + \sqrt{n\sigma^2\left[\ln\ln\right]_+(n\sigma^2) + n\sigma^2\ln\frac{2}{\delta}}\right)\right)^2$$

$$= \frac{D_0^2}{n}\left(1 + \ln\frac{2}{\delta}\right)^2 + D_0^2\sigma^2\left[\ln\ln\right]_+(n\sigma^2) + D_0^2\sigma^2\ln\frac{2}{\delta}$$

$$+ 2D_0^2\left(1 + \ln\frac{2}{\delta}\right)\sqrt{\frac{\sigma^2\left[\ln\ln\right]_+(n\sigma^2) + \sigma^2\ln\frac{2}{\delta}}{n}}$$

$$\leq K_1\left(1 + \ln\frac{1}{\delta} + \left[\ln\ln\right]_+(n\sigma^2)\right)$$

for some absolute constant $K_1$. The last inequality follows by $n \geq \ln\frac{1}{\delta}$.

Thus, by a union bound, with probability at least $1 - \delta$, for all $n$, $n\sigma^2 \leq K_0 V_n + K_0(K_1 + 2)\ln\frac{1}{\delta} + K_0 K_1\left[\ln\ln\right]_+(n\sigma^2) + K_0(K_1 + 3)$.

Let $K_2 > 0$ be an absolute constant such that $\forall x \geq K_2$, $K_0 K_1\left[\ln\ln\right]_+ x \leq \frac{x}{2}$.

Now if $n\sigma^2 \geq K_2$, then $n\sigma^2 \leq K_0 V_n + K_0(K_1 + 2)\ln\frac{1}{\delta} + \frac{n\sigma^2}{2} + K_0(K_1 + 3)$, and thus

$$n\sigma^2 \leq 2K_0 V_n + 2K_0(K_1 + 2)\ln\frac{1}{\delta} + 2K_0(K_1 + 3) + K_2 \tag{7}$$

If $n\sigma^2 \leq K_2$, clearly (7) holds. This concludes the proof. $\qquad\square$

We note that Proposition 2 is immediate by applying above lemma to Lemma 6.

**Lemma 9.** *Take any $\delta, n > 0$. Then with probability at least $1 - \delta$,*

$$V_n \leq 4n\sigma^2 + 8\ln\frac{1}{\delta}$$

*Proof.* Applying Bernstein's Inequality to $Y_i^2$, and noting that $\mathrm{Var}(Y_i^2) \leq 4\sigma^2$ since $|Y_i| \leq 2$, we have with probability at least $1 - \delta$,

$$\sum_{i=1}^{n} Y_i^2 \leq \frac{4}{3}\ln\frac{1}{\delta} + n\sigma^2 + \sqrt{8n\sigma^2\ln\frac{1}{\delta}}$$

$$\leq 4\ln\frac{1}{\delta} + 2n\sigma^2$$

The last inequality follows by the fact that $\sqrt{4ab} \leq a + b$.

The desired result follows by noting that $V_n = \frac{n}{n-1}\left(\sum_{i=1}^{n} Y_i^2 - \frac{1}{n}\left(\sum_{i=1}^{n} Y_i\right)^2\right) \leq 2\sum_{i=1}^{n} Y_i^2$.
$\qquad\square$

### C.2 Bounds of distances among probability distributions

**Lemma 10.** *If $P, Q$ are two probability distributions on a countable support $\mathcal{X}$, then*

$$d_{KL}(P \parallel Q) \leq \sum_x \frac{(P(x) - Q(x))^2}{Q(x)}$$

*Proof.*

$$
\begin{aligned}
d_{\mathrm{KL}}\left(P \parallel Q\right) & = \sum_{x} P(x) \ln \frac{P(x)}{Q(x)} \\
& \leq \sum_{x} P(x) \left(\frac{P(x)}{Q(x)} - 1\right) \\
& = \sum_{x} \frac{(P(x) - Q(x))^2}{Q(x)}
\end{aligned}
$$

The first inequality follows by $\ln x \leq x - 1$. The second equality follows by $\sum_{x} P(x) \left(\frac{P(x)}{Q(x)} - 1\right) = \sum_{x} \left(\frac{P^2(x) - P(x)Q(x)}{Q(x)} - P(x) + Q(x)\right) = \sum_{x} \frac{(P(x) - Q(x))^2}{Q(x)}$. $\qquad\square$

Define

$$
\begin{aligned}
P_0\left(Y = \perp \mid x\right) & = 1 - \left|x - \frac{1}{2}\right|^{\alpha} \\
P_0\left(Y = 0 \mid x\right) & = \begin{cases} \left(x - \frac{1}{2}\right)^{\alpha}\left(1 - \left(x - \frac{1}{2}\right)^{\beta}\right)/2 & x > \frac{1}{2} \\ \left(\frac{1}{2} - x\right)^{\alpha}\left(1 + \left(\frac{1}{2} - x\right)^{\beta}\right)/2 & x \leq \frac{1}{2} \end{cases} \\
P_0\left(Y = 1 \mid x\right) & = \begin{cases} \left(x - \frac{1}{2}\right)^{\alpha}\left(1 + \left(x - \frac{1}{2}\right)^{\beta}\right)/2 & x > \frac{1}{2} \\ \left(\frac{1}{2} - x\right)^{\alpha}\left(1 - \left(\frac{1}{2} - x\right)^{\beta}\right)/2 & x \leq \frac{1}{2} \end{cases}
\end{aligned}
$$

and

$$
\begin{aligned}
P_1\left(Y = \perp \mid x\right) & = 1 - \left|x - \epsilon - \frac{1}{2}\right|^{\alpha} \\
P_1\left(Y = 0 \mid x\right) & = \begin{cases} \left(x - \epsilon - \frac{1}{2}\right)^{\alpha}\left(1 - \left(x - \epsilon - \frac{1}{2}\right)^{\beta}\right)/2 & x > \epsilon + \frac{1}{2} \\ \left(\epsilon + \frac{1}{2} - x\right)^{\alpha}\left(1 + \left(\epsilon + \frac{1}{2} - x\right)^{\beta}\right)/2 & x \leq \epsilon + \frac{1}{2} \end{cases} \\
P_1\left(Y = 1 \mid x\right) & = \begin{cases} \left(x - \epsilon - \frac{1}{2}\right)^{\alpha}\left(1 + \left(x - \epsilon - \frac{1}{2}\right)^{\beta}\right)/2 & x > \epsilon + \frac{1}{2} \\ \left(\epsilon + \frac{1}{2} - x\right)^{\alpha}\left(1 - \left(\epsilon + \frac{1}{2} - x\right)^{\beta}\right)/2 & x \leq \epsilon + \frac{1}{2} \end{cases}
\end{aligned}
$$

**Lemma 11.** *Let $P_0$, $P_1$ be the distributions defined above. If $x \in [0,1]$, $\epsilon \leq \min\left\{\left(\frac{1}{2}\right)^{1/\beta}, \left(\frac{4}{5}\right)^{1/\alpha}, \frac{1}{4}\right\}$, then*

$$
\sum_{y} \frac{\left(P_0(Y = y \mid x) - P_1(Y = y \mid x)\right)^2}{P_0(Y = y \mid x) + P_1(Y = y \mid x)} = O\left(\epsilon^{\alpha} + \epsilon^2\right) \tag{8}
$$

*Proof.* By symmetry, it suffices to show for $0 \leq x \leq \frac{1+\epsilon}{2}$. Let $t = \frac{1}{2} + \epsilon - x$.

We first show (8) holds for $\frac{\epsilon}{2} \leq t \leq \epsilon$ (i.e. $\frac{1}{2} \leq x \leq \frac{1+\epsilon}{2}$).

We claim $\min_{y}\left(P_0(Y = y \mid X = t) + P_1(Y = y \mid X = t)\right) \geq \frac{1}{2}\left(\frac{\epsilon}{2}\right)^{\alpha}$. This is because:

- $P_0(Y = \perp \mid X = t) + P_1(Y = \perp \mid X = t) = 1 - (\epsilon - t)^{\alpha} + 1 - t^{\alpha} \geq 2 - 2\epsilon^{\alpha} \geq \frac{1}{2}\left(\frac{\epsilon}{2}\right)^{\alpha}$ where the last inequality follows by $\epsilon \leq \left(\frac{4}{5}\right)^{1/\alpha}$;

- $2\left(P_0(Y = 0 \mid X = t) + P_1(Y = 0 \mid X = t)\right) = (\epsilon - t)^{\alpha}\left(1 - (\epsilon - t)^{\beta}\right) + t^{\alpha}\left(1 + t^{\beta}\right) \geq t^{\alpha}\left(1 + t^{\beta}\right) \geq \left(\frac{\epsilon}{2}\right)^{\alpha}$. Therefore, $P_0(Y = 0 \mid X = t) + P_1(Y = 0 \mid X = t) \geq \frac{1}{2}\left(\frac{\epsilon}{2}\right)^{\alpha}$.

- Similarly, $P_0(Y = 1|X = t) + P_1(Y = 1|X = t) \geq \frac{1}{2} \left(\frac{\epsilon}{2}\right)^\alpha$.

Besides,

$$\sum_y \left(P_0(Y = y|X = t) - P_1(Y = y|X = t)\right)^2$$

$$= (t^\alpha - (\epsilon - t)^\alpha)^2 + \frac{1}{4} \left(t^\alpha \left(1 - t^\beta\right) - (\epsilon - t)^\alpha \left(1 + (\epsilon - t)^\beta\right)\right)^2$$

$$+ \frac{1}{4} \left(t^\alpha \left(1 + t^\beta\right) - (\epsilon - t)^\alpha \left(1 - (\epsilon - t)^\beta\right)\right)^2$$

$$= (t^\alpha - (\epsilon - t)^\alpha)^2 + \frac{1}{4} \left(t^\alpha - (\epsilon - t)^\alpha - t^{\alpha+\beta} - (\epsilon - t)^{\alpha+\beta}\right)^2$$

$$+ \frac{1}{4} \left(t^\alpha - (\epsilon - t)^\alpha + t^{\alpha+\beta} + (\epsilon - t)^{\alpha+\beta}\right)^2$$

$$\overset{(a)}{\leq} (t^\alpha - (\epsilon - t)^\alpha)^2 + \frac{1}{2} (t^\alpha - (\epsilon - t)^\alpha)^2 + \frac{1}{2} \left(t^{\alpha+\beta} + (\epsilon - t)^{\alpha+\beta}\right)^2$$

$$+ \frac{1}{2} (t^\alpha - (\epsilon - t)^\alpha)^2 + \frac{1}{2} \left(t^{\alpha+\beta} + (\epsilon - t)^{\alpha+\beta}\right)^2$$

$$= 2 (t^\alpha - (\epsilon - t)^\alpha)^2 + \left(t^{\alpha+\beta} + (\epsilon - t)^{\alpha+\beta}\right)^2$$

$$\leq 2\epsilon^{2\alpha} + 4\epsilon^{2\alpha+2\beta}$$

$$\leq 6\epsilon^{2\alpha}$$

where (a) follows by the inequality $(a + b)^2 \leq 2a^2 + 2b^2$ for any $a, b$.

Therefore, we get $\sum_y \frac{(P_0(Y=y|x) - P_1(Y=y|x))^2}{P_0(Y=y|x) + P_1(Y=y|x)} \leq \frac{\sum_y (P_0(Y=y|x) - P_1(Y=y|x))^2}{\min_y (P_0(Y=y|x) + P_1(Y=y|x))} \leq 12 * 2^\alpha \epsilon^\alpha$ when $\frac{1}{2} \leq x \leq \frac{1+\epsilon}{2}$.

Next, We show (8) holds for $\epsilon \leq t \leq \frac{1}{2} + \epsilon$ (i.e. $0 \leq x \leq \frac{1}{2}$). We will show $\frac{(P_0(Y=y|x) - P_1(Y=y|x))^2}{P_0(Y=y|x) + P_1(Y=y|x)} = O\left(\epsilon^\alpha + \epsilon^2\right)$ for $Y = \perp, 1, 0$.

For $Y = \perp$, for the denominator,

$$P_0(Y = \perp |X = t) + P_1(Y = \perp |X = t) = 2 - t^\alpha - (t - \epsilon)^\alpha \geq 2 - \left(\frac{3}{4}\right)^\alpha - \left(\frac{1}{2}\right)^\alpha$$

For the numerator,

$$(P_0(Y = \perp |X = t) - P_1(Y = \perp |X = t))^2 = (t^\alpha - (t - \epsilon)^\alpha)^2 = t^{2\alpha} \left(1 - \left(1 - \frac{\epsilon}{t}\right)^\alpha\right)^2$$

By Lemma 13, if $\alpha \geq 1$, $t^{2\alpha} \left(1 - \left(1 - \frac{\epsilon}{t}\right)^\alpha\right)^2 \leq t^{2\alpha} \left(\alpha \frac{\epsilon}{t}\right)^2 = t^{2\alpha-2} (\alpha\epsilon)^2 = O\left(\epsilon^2\right)$. If $0 \leq \alpha \leq 1$, $t^{2\alpha} \left(1 - \left(1 - \frac{\epsilon}{t}\right)^\alpha\right)^2 \leq t^{2\alpha} \left(\frac{\epsilon}{t}\right)^2 = t^{2\alpha-2}\epsilon^2 \leq \epsilon^{2\alpha}$.

Thus, we have $\frac{(P_0(Y=\perp|x) - P_1(Y=\perp|x))^2}{P_0(Y=\perp|x) + P_1(Y=\perp|x)} = O\left(\epsilon^{2\alpha} + \epsilon^2\right)$.

For $Y = 1$, for the denominator,

$$2 \left(P_0(Y = 1|X = t) + P_1(Y = 1|X = t)\right) = t^\alpha \left(1 - t^\beta\right) + (t - \epsilon)^\alpha \left(1 - (t - \epsilon)^\beta\right)$$

$$\geq t^\alpha \left(1 - t^\beta\right)$$

$$\geq t^\alpha \left(1 - \left(\frac{3}{4}\right)^\beta\right)$$

For the numerator,

$$(P_0(Y = 1|X = t) - P_1(Y = 1|X = t))^2$$

$$= \frac{1}{4}\left(t^\alpha \left(1 - t^\beta\right) - (t - \epsilon)^\alpha \left(1 - (t - \epsilon)^\beta\right)\right)^2$$

$$\leq \frac{1}{2}\left(t^\alpha - (t - \epsilon)^\alpha\right)^2 + \frac{1}{2}\left(t^{\alpha+\beta} - (t - \epsilon)^{\alpha+\beta}\right)^2$$

$$= \frac{1}{2}t^{2\alpha}\left(1 - (1 - \tfrac{\epsilon}{t})^\alpha\right)^2 + \frac{1}{2}t^{2\alpha+2\beta}\left(1 - (1 - \tfrac{\epsilon}{t})^{\alpha+\beta}\right)^2$$

$$\leq \frac{1}{2}t^{2\alpha}\left(1 - (1 - \tfrac{\epsilon}{t})^\alpha\right)^2 + \frac{1}{2}t^{2\alpha}\left(1 - (1 - \tfrac{\epsilon}{t})^{\alpha+\beta}\right)^2$$

If $\alpha \geq 1$, by Lemma 13, $\frac{1}{2}t^{2\alpha}\left(1 - (1 - \tfrac{\epsilon}{t})^\alpha\right)^2 + \frac{1}{2}t^{2\alpha}\left(1 - (1 - \tfrac{\epsilon}{t})^{\alpha+\beta}\right)^2 \leq \frac{1}{2}t^{2\alpha}\left(\alpha\tfrac{\epsilon}{t}\right)^2 + \frac{1}{2}t^{2\alpha}\left((\alpha + \beta)\tfrac{\epsilon}{t}\right)^2 = \left(\frac{1}{2}\alpha^2 + \frac{1}{2}(\alpha + \beta)^2\right)t^{2\alpha-2}\epsilon^2$. Thus, $\frac{(P_0(Y=1|x)-P_1(Y=1|x))^2}{P_0(Y=1|x)+P_1(Y=1|x)} \leq \left(\frac{1}{2}\alpha^2 + \frac{1}{2}(\alpha + \beta)^2\right)t^{\alpha-2}\epsilon^2/\left(1 - \left(\tfrac{3}{4}\right)^\beta\right)$ which is $O(\epsilon^2)$ if $\alpha \geq 2$ and $O(\epsilon^\alpha)$ if $\alpha \leq 2$.

If $\alpha \leq 1$ and $\alpha + \beta \geq 1$, by Lemma 13, $\frac{1}{2}t^{2\alpha}\left(1 - (1 - \tfrac{\epsilon}{t})^\alpha\right)^2 + \frac{1}{2}t^{2\alpha}\left(1 - (1 - \tfrac{\epsilon}{t})^{\alpha+\beta}\right)^2 \leq \frac{1}{2}t^{2\alpha}\left(\tfrac{\epsilon}{t}\right)^2 + \frac{1}{2}t^{2\alpha}\left((\alpha + \beta)\tfrac{\epsilon}{t}\right)^2 = \left(\frac{1}{2} + \frac{1}{2}(\alpha + \beta)^2\right)t^{2\alpha-2}\epsilon^2 \leq \left(\frac{1}{2} + \frac{1}{2}(\alpha + \beta)^2\right)t^{2\alpha-2}\epsilon^2$. Thus, $\frac{(P_0(Y=1|x)-P_1(Y=1|x))^2}{P_0(Y=1|x)+P_1(Y=1|x)} \leq \left(\frac{1}{2} + \frac{1}{2}(\alpha + \beta)^2\right)t^{\alpha-2}\epsilon^2/\left(1 - \left(\tfrac{3}{4}\right)^\beta\right) = O(\epsilon^\alpha)$.

If $\alpha \leq 1, \alpha+\beta \leq 1$, by Lemma 13, $\frac{1}{2}t^{2\alpha}\left(1 - (1 - \tfrac{\epsilon}{t})^\alpha\right)^2 + \frac{1}{2}t^{2\alpha}\left(1 - (1 - \tfrac{\epsilon}{t})^{\alpha+\beta}\right)^2 \leq \frac{1}{2}t^{2\alpha}\left(\tfrac{\epsilon}{t}\right)^2 + \frac{1}{2}t^{2\alpha}\left(\tfrac{\epsilon}{t}\right)^2 = t^{2\alpha-2}\epsilon^2$. Thus, $\frac{(P_0(Y=1|x)-P_1(Y=1|x))^2}{P_0(Y=1|x)+P_1(Y=1|x)} \leq t^{\alpha-2}\epsilon^2/\left(1 - \left(\tfrac{3}{4}\right)^\beta\right) = O(\epsilon^\alpha)$.

Therefore, we have $\frac{(P_0(Y=1|x)-P_1(Y=1|x))^2}{P_0(Y=1|x)+P_1(Y=1|x)} = O\left(\epsilon^\alpha + \epsilon^2\right)$.

Likewise, we can get $\frac{(P_0(Y=0|x)-P_1(Y=0|x))^2}{P_0(Y=0|x)+P_1(Y=0|x)} = O\left(\epsilon^\alpha + \epsilon^2\right)$. So we prove $\sum_y \frac{(P_0(Y=y|x)-P_1(Y=y|x))^2}{P_0(Y=y|x)+P_1(Y=y|x)} = O\left(\epsilon^\alpha + \epsilon^2\right)$ when $x \leq \frac{1}{2}$. This concludes the proof. $\square$

### C.3 Other lemmas

**Lemma 12.** *([20], Lemma 4) For sufficiently large $d > 0$, there is a subset $M \subset \{0,1\}^d$ with following properties: (i) $|M| \geq 2^{d/48}$; (ii) $\|v - v'\|_0 > \frac{d}{12}$ for any two distinct $v, v' \in M$; (iii) for any $i = 1, \ldots, d$, $\frac{1}{24} \leq \frac{1}{M}\sum_{v \in M} v_i \leq \frac{3}{24}$.*

**Lemma 13.** *If $x \leq 1, r \geq 1$, then $(1 - x)^r \geq 1 - rx$ and $1 - (1 - x)^r \leq rx$.*

*If $0 \leq x \leq 1, 0 \leq r \leq 1$, then $(1 - x)^r \geq \frac{1-x}{1-x+rx}$ and $1 - (1 - x)^r \leq \frac{rx}{1-(1-r)x} \leq x$.*

Inequalities above are know as Bernoulli's inequalities. One proof can be found in [16].

**Lemma 14.** *Suppose $\epsilon, \tau$ are positive numbers and $\delta \leq \frac{1}{2}$. Suppose $\{Z_i\}_{i=1}^\infty$ is a sequence of i.i.d random variables bounded by 1, $\mathbb{E}Z_i \geq \tau\epsilon$, and $\mathrm{Var}(Z_i) = \sigma^2 \leq 2\epsilon$. Define $V_n = \frac{n}{n-1}\left(\sum_{i=1}^n Z_i - \frac{1}{n}\left(\sum_{i=1}^n Z_i\right)^2\right)$, $q_n = q(n, V_n, \delta)$ as Procedure 2. If $n \geq \frac{\eta}{\tau\epsilon}\ln\frac{1}{\delta}$ for some sufficiently large number $\eta$ (to be specified in the proof), then with probability at least $1 - \delta$, $\frac{q_n}{n} - \mathbb{E}Z_i \leq -\tau\epsilon/2$.*

*Proof.* By Lemma 9, with probability at least $1 - \delta$, $V_n \leq 4n\sigma^2 + 8\ln\frac{1}{\delta}$, which implies

$$q_n \leq D_1\left(1 + \ln\frac{1}{\delta} + \sqrt{\left(4n\sigma^2 + 9\ln\frac{1}{\delta} + 1\right)\left([\ln\ln]_+ \left(4n\sigma^2 + 9\ln\frac{1}{\delta} + 1\right) + \ln\frac{1}{\delta}\right)}\right)$$

We denote the RHS by $q$.

On this event, we have

$$\begin{aligned}
\frac{q_n}{n} - \mathbb{E}Z_i \;\;\leq\;\; & \frac{q}{n} - \tau\epsilon \\
=\;\; & \tau\epsilon\left(\frac{q}{n\tau\epsilon} - 1\right) \\
\overset{(a)}{\leq}\;\; & \tau\epsilon\left(\frac{2D_1}{\eta} + \frac{D_1}{\eta\ln\frac{1}{\delta}}\sqrt{\frac{9\eta}{\tau}\ln\frac{1}{\delta}\left([\ln\ln]_+\,(\frac{9\eta}{\tau}\ln\frac{1}{\delta}) + \ln\frac{1}{\delta}\right)} - 1\right) \\
=\;\; & \tau\epsilon\left(\frac{2D_1}{\eta} + D_1\sqrt{\frac{9}{\eta\tau\ln\frac{1}{\delta}}[\ln\ln]_+\,(\frac{9\eta}{\tau}\ln\frac{1}{\delta}) + \frac{9}{\eta\tau}} - 1\right)
\end{aligned}$$

where (a) follows from $\frac{q}{n}$ being monotonically decreasing with respect to $n$. By choosing $\eta$ sufficiently large, we have $\frac{2D_1}{\eta} + D_1\sqrt{\frac{9}{\eta\tau\ln\frac{1}{\delta}}[\ln\ln]_+\,(\frac{9\eta}{\tau}\ln\frac{1}{\delta}) + \frac{9}{\eta\tau}} - 1 \leq -\frac{1}{2}$, and thus $\frac{q_n}{n} - \mathbb{E}Z_i \leq -\tau\epsilon/2$.

$\square$

## Footnotes

[2] Actually we can use Le Cam's method to prove this one dimensional case (which only needs to construct 2 distributions instead of 4 here), but this proof can be generalized to the multidimensional case more easily.