[Reviews · NeurIPS 2016]

Reviewer 1

Summary

This article presents a method for adaptive learning in the environment with labellers exhibiting certain amount of noise and uncertainty. The proposed method tries to optimally use the labellers in such a context and provide an adaptive approach in utilizing the questions to the labellers.

Qualitative Assessment

Are you sure that this is the best presentation of your research results? The article is properly motivated, but the exposition of the material is somewhat messy and lacks structure. Some of the formalisms are not properly explained in text. Some of the introduced symbols are not explained. Some of the used constants are not motivated. What does it mean that something was empirically estimated? Line 173: why is the interval chosen to be 3/4? The article does not have proper evaluation section. Please provide evidence and make the evaluation detailed enough so it is reproducible by others.

Confidence in this Review

2-Confident (read it all; understood it all reasonably well)


Reviewer 2

Summary

This paper studies the problem of active learning under a labeler that provides noisy labels as well as abstains when the examples are close to the decision boundary. The specific setting studies is the MQ model where the learner can query for the label of any example of its choice. For an example that is delta close to the boundary, the labeler can abstain with probability (1-delta^\alpha) and provides a noisy label with probability .5(1-delta^\beta). The contribution of the paper is two folds: 1. Generalize previous results on this model by providing an algorithm that does not need prior knowledge of the noisy rate and the abstention rate. 2. Generalize previous results on threshold function to higher dimensional smooth deacon boundaries. In both cases, the paper provides near optimal sample complexity.

Qualitative Assessment

this is a nice contribution. The knowledge of model parameters is an annoying aspect of many algorithms and the paper get away with it in a recently introduced model. The techniques used are straightforward and are similar to some previous work in active learning that also deals with unknown parameters. However, I think there is enough technical depth in the paper for NIPS.

Confidence in this Review

2-Confident (read it all; understood it all reasonably well)


Reviewer 3

Summary

The paper studies active learning where the labelers may make errors and abstain from providing a label. The paper shows lower bounds and upper bounds on the number of queries needed in the one dimensional case and d dimensional case. One possible motivation for this work is the results of Shah and Zhou “Double or Nothing: Multiplicative Incentive Mechanisms for Crowdsourcing” that shows that the only incentive compatible mechanism (under some assumptions) for crowed sourcing requires that workers abstain when they are uncertain.

Qualitative Assessment

• You are making the assumption that the hypothesis is the epigraph of holder smooth function. How realistic is this assumption? For example, can the decision boundaries of neural network, decision trees and kernel models have these properties? • Line 120: you are making here the assumption that the distribution is uniform over the unit cube. This should be made explicitly. How sensitive are the results to this assumptions? For example, Freund, Seung, Shamir and Tishby in their work on the query by committee algorithm used an assumption that the distribution is within a constant from uniform. • Assumption 3 seems very restrictive since it assumes equality. • Line 162 – can you show how does the assumption result in the decision boundary being a single point? • The technique uses membership queries and furthermore assume that the labeler is not consistent in the sense that when queried multiple times on the same point it may return different answers. Both assumptions are questionable. • Theorem 2: since the algorithm performs the significance test at every round, don’t you need to use the union bound to make sure that the significance test is true for all queries? In that case, the significance bound for the test should be updated to reflect the number of times it is being used. • Lines 232-234: you claim that errors are more harmful than missing labels. Is it a universal truth or specific to the current set of assumptions? • Theorem 9: the number of queries made is exponential in the dimension whereas standard PAC models are typically linear in the dimension. In what cases is it better to use your model? In their response, the authors address many of my concerns. The constraints are not "as bad" as I thought they are. Nevertheless, there are still many constraints in this setting and no demonstration of a case in which these constraints hold. I think this paper can be much improved and would create bigger impact after such improvements are implemented.

Confidence in this Review

2-Confident (read it all; understood it all reasonably well)


Reviewer 4

Summary

This paper focuses on active learning when the labeler will return the incorrect labels and provides some theoretical results on the sample complexity of active learning.

Qualitative Assessment

This paper is not well written. Most space of this paper is used to state the algorithm and analysis for 1 dimensional case. However, the multidimensional case is the most important part in this paper. There is no intuition or explanation about every Assumption and Theorem in this paper, it is hard to understand why these Assumptions are necessary for active learning to save queries and what insight the Theorems will bring into active learning. There are too many assumptions in this paper, e.g., holder smooth, c-growth, smooth boundary fragment, Assumptions 1, 2 and 3. It is hard to understand, e.g., why the flat function is not good? Even if the ground truth function is a flat function? Does there exist an application which satisfies all these assumptions? In this paper, the authors assume that the noise rate is smaller than 1/2 and provide the results on the sample complexity. It is well-known that the sample complexity for unbounded Tsybakov noise is much larger than that for bounded Tsybakov noise. While in this paper the authors provide a general sample complexity for the noise rate smaller than 1/2. It is necessary to discuss the relationship between these results. What is f(\epsilon/2) in Theorems 2 and 3? I did not find it before Theorems 2 and 3.

Confidence in this Review

2-Confident (read it all; understood it all reasonably well)


Reviewer 5

Summary

This paper provides some theoretical analysis to active learning with imperfect labelers, which may assign noisy labels or abstain from labeling. Authors analyzed different noise and abstention models and present the number of queries required to learn a good classifier.

Qualitative Assessment

Active learning with imperfect labelers is a very practical setting. There are some studies for this problem, yet no theoretical analysis is given. So this paper is important contribution for this important task. It is an interesting observation that the query complexity of the abstention-only case is significantly smaller than the noise-only case. And this result may provide some insightful suggestions to future algorithm design of active learning. I did not check every step of the proofs. But the results are reasonable and significant. Especially given that this is a less studied yet important topic. I suggest the authors to perform some empirical study to further validate the theoretical results.

Confidence in this Review

1-Less confident (might not have understood significant parts)


Reviewer 6

Summary

This theoretical work addresses active learning, in particular the query synthesis scenario with an abstaining and imperfect/noisy oracle for binary classification with smooth boundary fragments (as hypothesis class) in a d-dimensional feature space. It considers different noise and abstention models that are related to a variant of Tsybakov noise condition. It contributes an adaptive algorithm that requires less queries with a more informative oracle, and provides bounds for its query complexity. This algorithm does not require knowledge of the parameters of the noise or abstention models. In the case of a strictly monotonically increasing abstention rate of the oracle towards the true decision boundary (defined by the authors as a so-called c-growth property), the algorithm exploits the increasing abstention rate close to the decision boundary. This is a theoretical work, no experimental evaluation is provided. The first section provides an introduction and a review of the related work. This is followed by a specification of the active learner's context in section two, providing definitions of e.g. the c-growth property. In the third section, the active learning algorithm and its bounds are derived for a one dimensional feature space, where the decision boundary corresponds to a threshold. This is then extended to multidimensional feature spaces in section four.

Qualitative Assessment

The paper addresses an important and challenging active learning scenario, for which it provides an algorithm with interesting theoretical properties. Potential impact and usefulness: (1) An experimental evaluation (or at least a discussion) indicating exemplary classifier techniques that are candidates for combining them with the proposed active learning algorithm would be helpful for assessing the usefulness of the paper. Clarity and presentation: (2) Lines 51-56 are a repetition of lines 40-41 on page 2 (3) Figure 1, page 4, shouldn't it read x_1 in $g(\bar{x})=(x_1-0.4)^2+0.1$?)? (4) The abstract should provide additional information, e.g. on the addressed active learning scenario (query synthesis, binary classification in a multidimensional feature space).

Confidence in this Review

1-Less confident (might not have understood significant parts)